

# Estimating NOx emissions of stack plumes using a high-resolution atmospheric chemistry model and satellite-derived NO2 columns

Maarten Krol[1,2], Bart van Stratum[1], Isidora Anglou[1,3], and Klaas Folkert Boersma[1,3]

[1]Meteorology and Air Quality, Wageningen University & Research, Wageningen, The Netherlands
[2]Institute for Marine and Atmospheric Research Utrecht (IMAU), Utrecht University, The Netherlands
[3]Royal Netherlands Meteorological Institute, De Bilt, The Netherlands
[1]Correspondence: Maarten Krol (maarten.krol@wur.nl)

**Abstract.** This work contributes to the European Monitoring and Verification Support (MVS) capacity for anthropogenic $CO_2$ emissions. Future satellite instruments that map $CO_2$ and $NO_2$ from space will focus on hot-spot emissions from cities and large point sources, where $CO_2$ emissions are accompanied by emissions of $NO_x$. To use $NO_x$ as proxy $CO_2$ emission, information about its atmospheric lifetime and the fraction of $NO_x$ present as $NO_2$ is required to interpret $NO_2$ plumes. This paper presents

Large Eddy Simulations with atmospheric chemistry of four large point sources world-wide. We find that the chemical evolution of the plumes depends strongly on the amount of $NO_x$ that is emitted, next to wind speed and direction. For large $NO_x$ emissions the chemistry is pushed in a high-$NO_x$ chemical regime over a length of almost 100 km downwind of the stack location. Other plumes with lower $NO_x$ emissions show a fast transition to an intermediate $NO_x$ chemical regime, with short $NO_x$ lifetimes. Simulated $NO_2$ columns mostly agree within 20% with the TROPOspheric Monitoring Instrument (TROPOMI),

signalling that the emissions used in the model were approximately correct. However, variability in the simulations is large, making a one-to-one comparison difficult. We find that wind speed variations should be accounted for in emission estimation methods. Moreover, results indicate that common assumptions about the $NO_2$ lifetime ($\approx$4 hours) and $NO_x$:$NO_2$ ratios ($\approx$1.3) in simplified methods that estimate emissions from $NO_2$ satellite data (e.g. Beirle et al., 2019) need revision.

## 1 Introduction

To monitor compliance to global carbon dioxide ($CO_2$) emission reduction targets, space-based remote sensing of total column $CO_2$ (XCO2) is thought to play a vital role in the future. The European Union funded project CoCO2 (https://coco2-project.eu/) aims to build prototype systems for an European Monitoring and Verification Support (MVS) capacity for anthropogenic $CO_2$ emissions. Since a large fraction of the anthropogenic $CO_2$ is emitted by point sources, CoCO2 specifically addresses the quantification of emissions from hot spots based on (upcoming) satellite data.

Indeed, it has been shown that satellites can detect $CO_2$ emissions of large stack emitters and cities using observations of e.g. OCO-2 and OCO-3 (Nassar et al., 2017; Hakkarainen et al., 2016, 2023; Zhang et al., 2023; Lin et al., 2023). Upcoming satellite missions, like the Copernicus CO2M mission (Pinty et al., 2019; Janssens-Maenhout et al., 2020; Sierk et al., 2019) will largely improve the spatial coverage of space-based XCO2 retrievals. But even with high spatially resolved XCO2 data, it remains challenging to derive emissions of point sources, which emit their $CO_2$ in a high and variable background that



is influenced by biosphere exchange and many diffuse sources in an urban environment. The target for the precision for an individual CO2M XCO2 sounding is therefore strict: 0.7 ppm, with an absolute bias of less than 0.5 ppm (Pinty et al., 2019). Moreover, the CO2M mission will be augmented with an instrument that simultaneously detects nitrogen dioxide columns (XNO2) (Kuhlmann et al., 2021). Depending on the technology implemented at the stack, $NO_x$ (NO + $NO_2$) is emitted in substantial quantities alongside $CO_2$. Since the atmospheric lifetime of $NO_2$ is rather short (in the order of 4 hours (Kuhlmann

et al., 2021; Hakkarainen et al., 2021)), the $NO_2$ background is much smaller compared to the $CO_2$ background, and plumes can be readily detected. Thus, XNO2 plumes can be used to filter XCO2 images, improving the emission quantification from point sources. Moreover, if the $NO_x$:$CO_2$ emission ratio is known, $NO_x$ emissions can be derived from XNO2 observations that can then be converted to $CO_2$ emissions using the emission ratio (Hakkarainen et al., 2021; Zhang et al., 2023).

    The TROPOspheric Monitoring Instrument (TROPOMI) on board the Copernicus Sentinel-5 Precursor satellite samples

XNO2 at a resolution of 5.5 km × 3.5 km at nadir (since August 2019, 7 km × 3.5 km before that date). This high spatial resolution of the TROPOMI XNO2 product offers the possibility to detect emissions from point sources and cities (Lorente et al., 2019; Beirle et al., 2021; Goldberg et al., 2019; Ialongo et al., 2021; Zhang et al., 2023). However, in contrast to $CO_2$, $NO_2$ is not chemically inert. To derive $NO_x$ emission from TROPOMI observations, a typical $NO_2$ lifetime of 4 hours is assumed (Kuhlmann et al., 2021). Moreover, the majority of the $NO_x$ emissions of power plants is emitted in the form of

nitrogen oxide (NO), which is converted to $NO_2$, mostly by reaction with ambient $O_3$. In the analysis of stack emissions, a $NO_x$ to $NO_2$ ratio of ≈ 1.3 is commonly assumed (Hakkarainen et al., 2021; Beirle et al., 2021). This value does not reflect the fact that $NO_x$ atmospheric chemistry is highly non-linear, with different chemical regimes depending on $NO_x$ mole fractions (Rohrer et al., 2014). To account for these non-linear effects in models, parameterizations have been developed (Vilà-Guerau de Arellano et al., 1990; Vinken et al., 2011; Wu et al., 2023) that need further testing.

Large uncertainties exist regarding the ability of atmospheric transport models to describe individual observed plumes (Brunner et al., 2023). Within the CoCO2 project, high resolution models are developed to simulate emissions from individual stacks. Here, we present results of 100 m resolution Large Eddy Simulations (LES) of four large point sources that emit substantial amounts of $NO_x$ and $CO_2$. Plumes of these facilities have been detected by space-borne instruments like TROPOMI (Beirle et al., 2021) and OCO-2 (Nassar et al., 2021). We will focus on the skill of our simulations to reproduce observed $NO_2$ plumes

from TROPOMI on individual days, by accounting for atmospheric chemistry. To this end, we embed our simulations within boundaries that are provided by Copernicus Atmospheric Monitoring Service (CAMS) for composition, and the Copernicus Climate Change Service (C3S) ERA5 data for meteorology.

    By analysing LES results of four large point sources we will address the following questions:

    – How does atmospheric chemistry affect the $NO_x$ plume?

– What is the impact of meteorology on plume dispersion?

    – How do the simulations compare to TROPOMI $NO_2$ observations?

    – What are the main factors that influence emission quantification from satellite observations?





The latter question links to ongoing efforts to use simplified models (Kuhlmann et al., 2021) to derive emissions from current satellite instruments, and is a core question in building an operational MVS system.

The paper is organised as follows: In Section 2 we present the chemistry scheme that has been implemented in the LES model, the cases that have been simulated, and the TROPOMI observations that are used for evaluating the simulations. In Section 3.1 we present the simulated meteorology, in Section 3.2 we analyse the simulated $NO_x$ chemistry in the plume, and in Section 3.3 we compare to TROPOMI XNO2 columns. Finally, in Section 4 we discuss the results and present the main conclusions.

## 2    Method

### 2.1    MicroHH

Simulations described here have been performed using the Large Eddy Simulation (LES) model MicroHH (van Heerwaarden et al., 2017). The LES implementation of MicroHH uses a surface model that is constrained to rough surfaces and high Reynolds numbers, which is a typical configuration for atmospheric flows. This model computes the surface fluxes of the

horizontal momentum components and the scalars (including thermodynamic variables) using Monin–Obukhov Similarity Theory (MOST) (Wyngaard, 2010). To parameterise the anisotropic subfilter-scale kinematic momentum flux tensor, MicroHH uses the Smagorinsky–Lilly model (Lilly, 1996; van Heerwaarden et al., 2017). For our simulations, we use a domain size of 50–100 km with grid cells of 100 m × 100 m in the horizontal and 25 m in the vertical dimension. MicroHH uses an adaptive time step depending on the local flow conditions (van Heerwaarden et al., 2017) that typically amounts to 1–5 s in the current

simulations. The emission of scalars from point and line sources is described in Ražnjević et al. (2022a) and Ražnjević et al. (2022b). The coupling of MicroHH with meteorological reanalysis data from ERA5 (Hersbach et al., 2020) using the open source python package named "the Large-eddy simulation and Single-column model—Large-Scale Dynamics ((LS)$^2$D)" is described in van Stratum et al. (2023). The simulations will focus a point source (stack) within a domain. Next to $CO_2$, the stack emits prescribed amounts of $NO_x$ and other pollutants. These latter species are involved in atmospheric chemistry. The

next section describes the implementation of atmospheric chemistry in MicroHH.

### 2.1.1    Atmospheric Chemistry Scheme

We devised a condensed chemistry scheme based on the scheme implemented in the Integrated Forecasting System (IFS) of European Centre for Medium-Range Weather Forecasts (ECMWF) and used for the Copernicus Atmosphere Monitoring Service (CAMS) reanalysis (Inness et al., 2019). The CB05 mechanism of the IFS chemistry is based on the version implemented

in the TM5 model (Huijnen et al., 2010). CB05 describes tropospheric chemistry with 55 species and 126 reactions. Since the residence time of air in the small LES model domain is relatively short (hours), the condensed scheme focuses on reproducing the $NO_x$ and $O_3$ chemistry of the full IFS scheme. We put less emphasis on the involved oxidation scheme of non-methane



hydrocarbons (NMHCs), and replace the relevant IFS species by one compound: R = propene = $C_3H_6$. Since we aim to compare to TROPOMI $NO_2$, we put extra emphasis on N-containing species.

Table 1 lists the species that are considered in MicroHH. The long-lived $CH_4$ and $H_2$ attain a fixed mole fraction in the domain. Other species are transported and/or emitted by the stack. The transported species are forced at the boundaries by information from the CAMS reanalysis. Here, $C_3H_6$ = R, ROOH, and $RO_2$ are lumped from IFS chemical compounds, as listed in Table 2.

The chemical reactions are generally taken from the IFS chemistry scheme and are listed in Table 3. Note that the reaction

scheme also considers surface deposition for $HNO_3$, $O_3$, NO, $NO_2$, HCHO, $H_2O_2$, and ROOH, as described in Visser (2022). For photolysis frequencies, we produced look-up tables using the TUV model (Madronich and Flocke, 1998). Here we took a simple approach using standard atmospheric profiles of aerosol and $O_3$. We evaluate the photolysis rates at 500 m above the surface with 15 minute time steps during a full diurnal cycle at the specific latitude, longitude, and day of the simulation.

In order to calibrate the reactions scheme to the IFS scheme, we employed a box model implementation of the reduced

scheme and compared this to the full IFS scheme. We performed two-day simulations with diurnal variation in radiation, representative for an atmospheric boundary layer, and considered two cases. The first case has high emissions of NO and no hydrocarbon emissions. In this case, results of the condensed scheme are nearly identical to those of the IFS scheme. Small differences are caused by the omission of e.g. $HNO_2$ in the condensed scheme. In the second case we additionally considered high emissions of hydrocarbons, represented by $C_3H_6$. Figure 1 shows the results for some main atmospheric species. Note that

we tuned the reaction products of reaction 27 (1.0 $RO_2$ + 1.5 HCHO) to obtain favorable comparisons for mainly $NO_2$. Results for HCHO deviate because the condensed scheme does not consider aldehydes, and produces formaldehyde instead (reaction 27). We consider the comparison with the full IFS scheme favorable and fit for purpose and proceed with a description of the numerical implementation of this chemistry scheme in MicroHH.



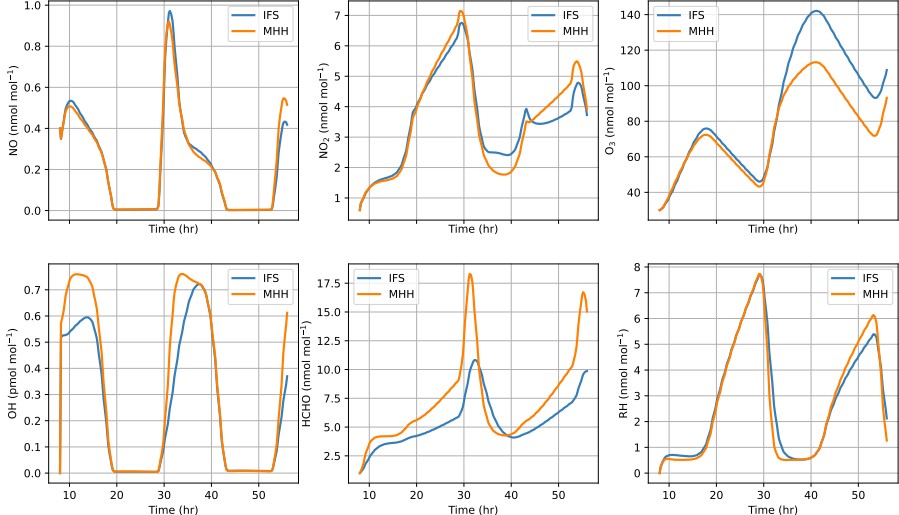

**Figure 1.** Box model comparison of the IFS scheme with the condensed MicroHH (MHH) scheme. Results of a two day simulation are shown with high emissions of NO and $C_3H_6$, starting at 8 AM. Time series are plotted for NO, $NO_2$, $O_3$, OH, HCHO, and RH ($C_3H_6$).

**Table 1.** Species simulated in the MicroHH model. Five compounds are emitted by the simulated stack, and six species are deposited at the surface (Visser, 2022). Status "–" indicates that only chemical sources and sinks are considered.

| Compound | Name | Status |
|---|---|---|
| $CH_4$ | Methane | Fixed (1800 nmol mol$^{-1}$) |
| $H_2$ | Hydrogen | Fixed (500 nmol mol$^{-1}$) |
| $O_3$ | Ozone | Deposited |
| NO | Nitrogen Oxide | Emitted and Deposited |
| $NO_2$ | Nitrogen Dioxide | Emitted and Deposited |
| $NO_3$ | Nitrate | – |
| $N_2O_5$ | Dinitrogen Pentoxide | – |
| $HNO_3$ | Nitric Acid | Deposited |
| OH· | Hydroxyl radical | – |
| $HO_2$· | Hydroxyperoxyl radical | – |
| $H_2O_2$ | Hydrogen Peroxide | Deposited |
| CO | Carbon Monoxide | Emitted |
| HCHO | Formaldehyde | Deposited |
| $CO_2$ | Carbon Dioxide | Inert and Emitted |
| $C_3H_6$ | Propene | Emitted (R) |
| $RO_2$· | Organic Peroxyl radical | – |
| ROOH | Organic Peroxide | Deposited |



**Table 2.** Lumping of IFS species into MicroHH tracers. The IFS chemistry scheme is described in Flemming et al. (2015).

| Compound | IFS species |
|---|---|
| $C_3H_6$ | PAR, $C_2H_4$, OLE, $C_5H_8$, $C_2H_5OH$, $C_3H_8$, $C_3H_6$, $C_{10}H_{16}$ |
| ROOH | ROOH, $CH_3OOH$ |
| $RO_2$ | $CH_3O_2$, $C_2O_3$, $ACO_2$, $IC_3H_7O_2$, $HYPROPO_2$ |

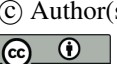

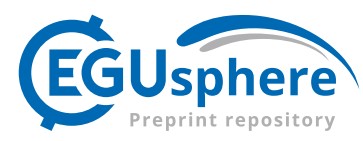

Table 3: Reaction scheme employed by MicroHH. M denotes air molecules, and $h\nu$ denotes radiation associated with photolysis. $C_M$ and $C_{H2O}$ are the air density and water vapor density, respectively, in (molecules cm$^{-3}$). TEMP is the temperature in K. The rate expressions refer to specific functions to evaluate the rate constants, and are taken from the IFS scheme.

| Reaction | Consumed | Produced | Rate |
|---|---|---|---|
| 0 | $O_3$ + OH | $HO_2$ | ARR3(1.7E-12, -940., TEMP) |
| 1 | $O_3$ + $HO_2$ | OH | ARR3(1.E-14, -490., TEMP) |
| 2 | OH + $HO_2$ | M | ARR3(4.8E-11, 250., TEMP) |
| 3 | $HO_2$ + $HO_2$ | $H_2O_2$ | EPR(3.E-13, 460., 2.1E-33, 920., 1.4E-21, 2200., $C_M$, $C_{H2O}$, TEMP) |
| 4 | $H_2O_2$ + OH | $HO_2$ | ARR3(2.9E-12, -160., TEMP) |
| 5 | $H_2$ + OH | $HO_2$ | ARR3(2.8E-12, -1800., TEMP) |
| 6 | NO + $O_3$ | $NO_2$ | ARR3(3.E-12, -1500., TEMP) |
| 7 | $NO_2$ + $O_3$ | $NO_3$ | ARR3(1.4E-13, -2470.,TEMP) |
| 8 | $NO_3$ + NO | 2 $NO_2$ | ARR3(1.8E-11, 110.,TEMP) |
| 9 | $NO_3$ + $NO_2$ +M | $N_2O_5$ +M | TROE$_{ifs}$(3.6E-30, 4.1, 1.9E-12, -0.2, 10., $C_M$, TEMP) |
| 10 | $N_2O_5$ +M | $NO_2$ + $NO_3$ | TROE$_{ifs2}$(1.3E-3, -3.5, 9.7E14, 0.1, 10., $C_M$, -11000., -11080., TEMP) |
| 11 | NO + $HO_2$ | OH + $NO_2$ | ARR3(3.3E-12, 270., TEMP) |
| 12 | $NO_2$ + OH | $HNO_3$ | TROE$_{no2oh}$(3.2E-30, 4.5, 3.E-11, 10, $C_M$, TEMP) |
| 13 | $NO_3$ + $HO_2$ | $HNO_3$ | 4.E-12 |
| 14 | $HNO_3$ + OH | $NO_3$ | RK28(2.4E-14, 460., 6.51E-34, 1335., 2.69E-17, 2199., $C_M$, TEMP) |
| 15 | $N_2O_5$ | 2 $HNO_3$ | 4e-4 |
| 16 | $CH_4$ + OH | $RO_2$ +h2o | ARR3(2.45E-12, -1775., TEMP) |
| 17 | $RO_2$ + $HO_2$ | ROOH | ARR3(3.8E-13, 780., TEMP) * (1. - (1. / (1. + ARR3(498., -1160., TEMP)))) |
| 18 | $RO_2$ + $HO_2$ | HCHO | ARR3(3.8E-13, 780., TEMP) * (1. / (1. + ARR3(498., -1160., TEMP))) |
| 19 | $RO_2$ + NO | HCHO + $HO_2$ + $NO_2$ | ARR3(2.8E-12, 300., TEMP) |
| 20 | $RO_2$ + $NO_3$ | HCHO + $NO_2$ + $HO_2$ | 1.2E-12 |
| 21 | ROOH + OH | 0.6 $RO_2$ + 0.4 HCHO + 0.4 OH | ARR3(3.8E-12, 200., TEMP) |
| 22 | HCHO + OH | CO + $HO_2$ | ARR3(5.5E-12, 125., TEMP) |
| 23 | HCHO + $NO_3$ | $HNO_3$ | 5.8E-16 |





**Table 3 – continued from previous page**

| Reaction | Consumed | Produced | Rate |
|---|---|---|---|
| 24 | $OH + CO$ | $HO_2$ | $TROE_{ccooh}(5.9E\text{-}33, 1.4, 1.1E\text{-}12, -1.3, 1.5E\text{-}13, -0.6, 2.1E9, -6.1, 0.6, C_M, TEMP)$ |
| 25 | $RO_2 + RO_2$ | $1.37\ HCHO + 0.74\ HO_2$ | $ARR3(9.5E\text{-}14, 390., TEMP)$ |
| 26 | $RH + O_3$ | $1.04\ HCHO + 0.19\ HO_2 +$ $0.33\ OH + 0.56\ CO + 0.31\ RO_2$ | $ARR3(5.5E\text{-}15,-1880.0,TEMP)$ |
| 27 | $RH + OH$ | $1.0\ RO_2 + 1.5\ HCHO$ | $k3rd_{iupac}(8.6E\text{-}27, 3.5, 3.E\text{-}11, 1., 0.6, C_M, 0.5, TEMP)$ |
| 28 | $RH + NO_3$ | $RO_2 + NO_2\ M$ | $ARR3(4.6E\text{-}13, -1155.,TEMP)$ |
| 29 | $O_3 + h\nu$ | $2\ OH$ | photolysis (+ branching) |
| 30 | $NO_2 + h\nu$ | $NO + O_3$ | photolysis |
| 31 | $N_2O_5 + h\nu$ | $NO_2 + NO_3$ | photolysis |
| 32 | $NO_3 + h\nu$ | $NO_2 + O_3$ | photolysis |
| 33 | $ROOH + h\nu$ | $HCHO + OH + HO_2$ | photolysis |
| 34 | $HCHO + h\nu$ | $CO$ | photolysis |
| 35 | $HCHO + h\nu$ | $CO + 2\ HO_2$ | photolysis |
| 36 | $H_2O_2 + h\nu$ | $2\ OH$ | photolysis |





### 2.1.2 Numerical Implementation

Tracers in MicroHH (van Heerwaarden et al., 2017) are advected using a second-order scheme with a fifth-order interpolation with an imposed flux limiter to ensure monotonicity. Time is advanced with a third-order Runga-Kutta scheme (RK3). During time integration, MicroHH collects tendencies (e.g. advection, cloud processes, surface exchange) of all meteorological variables and chemistry tracers. Tendencies for tracer emission, deposition and chemistry are added to these tendencies. Tendencies of chemistry and deposition are evaluated with code that is automatically generated in C-language by a Kinetic Pre-Processor

(KPP) (Damian et al., 2002). This code integrates the chemistry rate equations (including deposition terms) from time $t$ to $t + dt$ using the highly accurate Rosenbrock solver. An accurate solver for chemistry is required, because the chemistry rate equations can be very stiff due to the fast time-scales involved.

After this integration, which is performed for each of the three sub-steps of the RK3 scheme, tendencies of concentration $C$ are evaluated as:

$$\left[\frac{\partial C}{\partial t}\right]_{\text{chemistry}} = \frac{C(t+dt) - C(t)}{dt}. \tag{1}$$

The calculation of the chemistry tendencies is evaluated after the calculation of all other tendencies, including the emissions. The concentration $C(t)$ before the start of the chemistry integration is updated by these tendencies as:

$$C(t) = C(t) + dt \times \left[\frac{\partial C}{\partial t}\right]_{\text{other processes}}, \tag{2}$$

where $dt$ is the sub-timestep of the RK3 integration (van Heerwaarden et al., 2017). After evaluation, the tendencies are added

to the tendencies of the other processes in the main time-integration scheme in MicroHH (van Heerwaarden et al., 2017):

$$\frac{\partial C}{\partial t} = \left[\frac{\partial C}{\partial t}\right]_{\text{other processes}} + \left[\frac{\partial C}{\partial t}\right]_{\text{chemistry}} \tag{3}$$

Note that this approach leads to many calls of the Rosenbrock solver (3 calls per full time step), which makes the numerical integration of chemistry slow. We found, however, that compromises in the numerical integration lead to numerical instabilities that may lead to negative concentrations. In high-resolution simulations of large point sources, large spatial gradients will

occur, which are a likely cause of these numerical instabilities. Stack emissions are introduced in the model as described in Ražnjević et al. (2022b). To avoid numerical inaccuracies, point sources are emitted as 3D Gaussian functions that cover 4 grid-boxes in each dimension. Note that this leads to a slight "pre-dispersion" of point sources.

### 2.2 Simulated cases

One of the aims of the CoCO2 project (https://coco2-project.eu/) is to build a library of plumes. To that end, simulation

protocols have been designed (https://coco2-project.eu/sites/default/files/2021-07/CoCO2-D4.1-V1-0.pdf). Here, we present results of four cases listed in Table 4 that address emissions from point sources.

The Jänschwalde Power Station is a coal-fired power station near Cottbus, Germany, close to the German-Polish border. The Jänschwalde power station has 9 cooling towers (120 m high) in groups of three, of which only two towers per group are active. This facility has been studied in a couple of recent papers (Brunner et al., 2023; Kuhlmann et al., 2021; Beirle et al., 2021).



The Bełchatów Power Station is also a coal-fired power station near Bełchatów, in central Poland. Emissions are released from two 299 m high stacks. $CO_2$ emissions of this facility were addressed in Cusworth et al. (2021); Nassar et al. (2021).

     The Lipetsk steel plant is owned by the NLMK group, the largest steelmaker in Russia. This facility has been identified in earlier studies (Nassar et al., 2021; Reuter et al., 2019).

     Finally, the Matimba power station is a dry cooled, coal-fired power plant in the north-east of South Africa, approximately
300 km north of Johannesburg. The power plant has two 250 m high stacks. This case is based on Hakkarainen et al. (2021) and is also addressed in e.g. Hakkarainen et al. (2023); Reuter et al. (2019); Brunner et al. (2023).

     Emission details of these four cases are summarized in Table 5. These facilities are all major emitters of $CO_2$, with emission strengths ranging from 16-28 kmol $s^{-1}$. For chemical compounds, Matimba is clearly emitting more $NO_x$, while Lipetsk is a very strong emitter of CO.

**Table 4.** Simulation cases presented in this paper.

| CaseID | Facility | Simulation period |
|--------|----------|-------------------|
| JAE | Power plant Jänschwalde, Germany | 2018 May 22 + 23 |
| BEL | Power plant Bełchatów, Poland | 2018 June 6 + 7 |
| LIP | Steel plant Lipetsk, Russia | 2019 June 12 + 13 |
| MAT | Power plant Matimba, South Africa | 2020 July 24 + 25 |

**Table 5.** Emission configuration of the different simulations. Emissions are distributed vertically either as a probability density function (JAE, BEL), or as prescribed distribution. In the former case, emission height and $1\sigma$ values are given based on a plume rise calculation. In the latter case, we list the peak emission height and the percentage of the emissions emitted at that height. For JAE and BEL emissions are evenly distributed over the towers. Emission amounts and heights are taken from the modelling protocols (https://coco2-project.eu/sites/default/files/2021-07/CoCO2-D4.1-V1-0.pdf).

| CaseID | Lon | Lat | Height | $CO_2$ | $NO_2$ | NO | CO | $C_3H_6$ |
|--------|-----|-----|--------|--------|--------|----|----|----------|
| | ° | ° | m | kmol $s^{-1}$ | mol $s^{-1}$ | mol $s^{-1}$ | mol $s^{-1}$ | mol $s^{-1}$ |
| JAE 1 | 14.4622 | 51.8360 | 299.68±122.37 | 5.548 | 0.210 | 3.987 | 2.661 | 0.041 |
| JAE 2 | 14.4580 | 51.8361 | 299.68±122.37 | 5.548 | 0.210 | 3.987 | 2.661 | 0.041 |
| JAE 3 | 14.4538 | 51.8362 | 299.68±122.37 | 5.548 | 0.210 | 3.987 | 2.661 | 0.041 |
| JAE sum | | | | 16.644 | 0.629 | 11.960 | 7.984 | 0.122 |
| BEL 1 | 19.3285 | 51.2660 | 618.7±151.7 | 13.835 | 0.498 | 9.450 | 14.089 | 0.190 |
| BEL 2 | 19.3237 | 51.2660 | 618.7±151.7 | 13.835 | 0.498 | 9.450 | 14.089 | 0.190 |
| BEL sum | | | | 27.670 | 0.996 | 18.900 | 28.179 | 0.381 |
| LIP | 39.6296 | 52.5574 | 138 (75%) | 20.608 | 0.902 | 17.167 | 266.429 | 2.113 |
| MAT | 27.6106 | -23.668 | 300–425 (96%) | 18.044 | 2.139 | 40.637 | 2.271 | 0.238 |



Depending on the wind direction during the selected time periods and visual inspection of the TROPOMI NO$_2$ plumes, a modelling domain was set up around the point source. JAE, BEL, and LIP were modelled on a domain for 51.2 km × 51.2 km, while for the MAT case a domain of 102.4 km × 102.4 km was selected. All simulations employed a horizontal resolution of 100 m × 100 m. In the vertical, the domain size was 4000 m, with an equidistant grid of 25 m resolution. At the top of the domain, a buffer layer starting at 3250 m was used to damp gravity waves (van Heerwaarden et al., 2017). Radiative transfer

was calculated every 60 seconds using the RTE-RRTMGP radiative transfer model (Pincus et al., 2019). At the surface, we employed an interactive land surface model based on HTESSEL (Balsamo et al., 2011). We initialized our simulations using CAMS (composition) and ERA5 (meteorology) using LS$^2$D (van Stratum et al., 2023). During the simulations, boundaries were nudged towards time-varying profiles of CAMS and ERA5. For temperature, humidity, and momentum, circular boundary conditions were used. To avoid re-entering of emissions from the point source, we employed free outflow conditions for tracers

as described in Ražnjević et al. (2022b). Since the current focus is on stack emissions, surface fluxes of CO$_2$ and other tracers were ignored.

    Simulations were performed on the Dutch national supercomputer snellius, using 1024 cores (8 nodes). Typical run times of the simulations range from 2 days (JAE, BEL, LIP) to 5 days (MAT).

### 2.3    Observations

We compare the results of our simulations to TROPOMI satellite data. We downloaded level-2 TROPOMI NO$_2$ data from the Copernicus Open Access Hub (https://scihub.copernicus.eu). For consistency, we selected the product that was reprocessed (RPRO) with processor version 2.4.0. For JAE orbits 3136 (2018-05-22) and 3150 (2018-05-23) were downloaded; for BEL orbits 3349 (2018-06-06) and 3363 (2018-06-07); for LIP orbits 8611 (2019-06-12) and 8626 (2019-06-13); for MAT orbits 14402 (2020-07-24) and 14416 (2020-07-25). The latter two orbits provide data at nadir on 5.5 km × 3.5 km, while the nadir

resolution of the other orbits is 7 km × 3.5 km. The uncertainty in a single TROPOMI tropospheric column due to albedo, clouds, and aerosol amounts to 20–30% (Geffen et al., 2022; Riess et al., 2022).

    Figure 2 displays the tropospheric NO$_2$ columns in the TROPOMI product on the selected days and on the MicroHH modelling domains. Wind speed and direction as calculated with MicroHH are also shown in the panels. We selected only column retrievals with a qa value > 0.75.

First, the resolution of the TROPOMI product strongly depends on the satellite viewing angle. Second, for all cases, clear NO$_2$ plumes are visible. Only for the LIP case on 2019-06-12 the spread of the plume is limited due to low wind speeds. Moreover, many TROPOMI pixels are flagged on this day, likely due to aerosol and/or clouds. Finally, as expected and analysed later, the TROPOMI NO$_2$ columns depend strongly on the wind speed. A clear effect of wind speed is seen on the second day of the JAE case (2018-05-23), when columns are clearly reduced compared to the first day (2018-05-22).

In the further analysis of TROPOMI data, we will remove inconsistencies in the model–satellite comparison caused by the use of vertical NO$_2$ profiles from the coarse-grid TM5 model in the satellite product. This global chemistry transport model runs on a resolution of 1° × 1° and does not resolve the highly localized plume simulated by MicroHH. Since the sensitivity of the satellite measurement drops significantly for NO$_2$ that resides near the surface, mainly due to Rayleigh scattering, it is



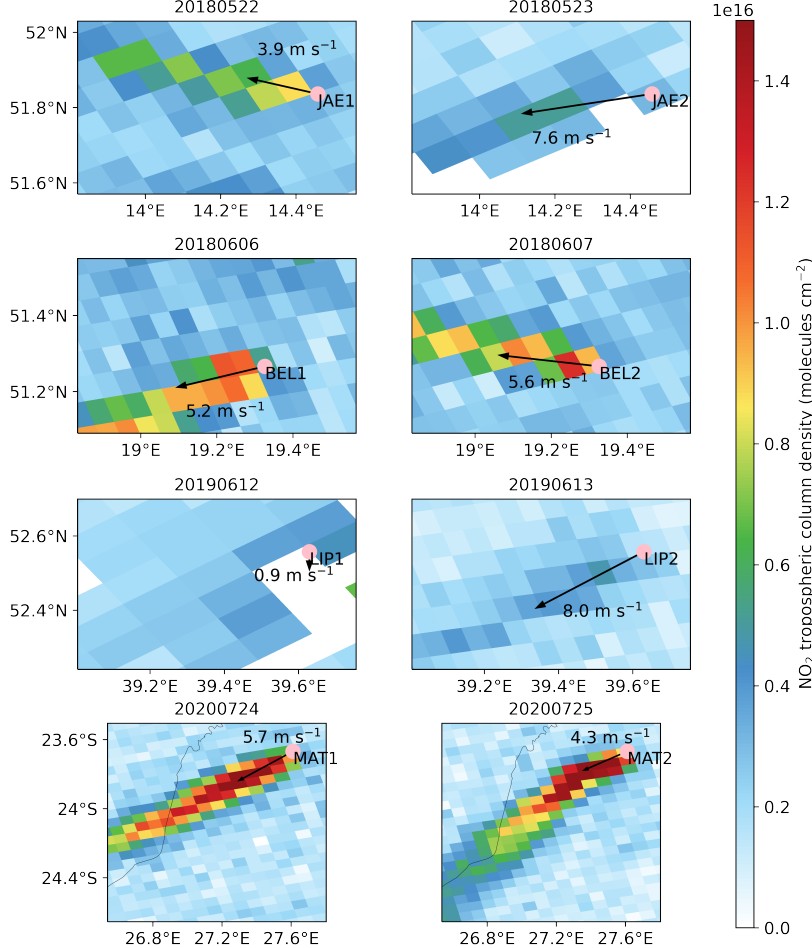

**Figure 2.** TROPOMI tropospheric $NO_2$ columns analysed in this paper. Per case, two days are considered with the first day in the left column (labeled 1) and the second day in the right column (labeled 2). White pixels refer to TROPOMI sounding with a qa value < 0.75. The central point of emission is labeled by the pink dot. The color scale is similar for all cases. Note that for the MAT case, the columns are substantially outside the color range and the considered domain is twice the size of the other cases. The arrows reflect the domain-averaged wind direction and speed calculated by MicroHH at TROPOMI overpass time. As will be discussed later, wind speed and direction are weighted in the vertical with the domain-average $NO_2$ profile.



important to correct for the differences in $NO_2$ profile shape and $NO_2$ amount between MicroHH and TM5. Previous studies
have shown strong impacts of the $NO_2$ profile and amount on satellite retrievals (Vinken et al., 2014; Visser et al., 2019).

The TM5 information about the vertical $NO_2$ distribution is stored in the TROPOMI data product in the form of a tropo-
spheric Averaging Kernel (AK) and Air Mass Factor (AMF). We employ the method outlined in Boersma et al. (2016) and
applied in Visser et al. (2019) for the OMI instrument. To this end, we sample the MicroHH $NO_2$ profile, augmented with
the CAMS profile above 4 km, on the pressure grid of the TM5-based tropospheric AK, and calculate a correction to the
tropospheric AMF as:

$$M_{trop,MHH} = M_{trop,TM5} \times \frac{\sum_{l=1}^{L} A_{trop,l} x_{l,MHH}}{\sum_{l=1}^{L} x_{l,MHH}}. \tag{4}$$

Here, $M_{trop}$ is the tropospheric AMF of the MicroHH model (MHH) or the TM5 model (stored in the satellite product),
$A_{trop,l}$ is tropospheric averaging kernel element for layer $l$ (also stored in the satellite product). $x_{l,MHH}$ is the modelled $NO_2$
column density sampled on the TM5 pressure grid, and L is the uppermost TM5 layer in the troposphere. In Section 3.3 we
will compare MicroHH tropospheric $NO_2$ columns to corrected and uncorrected TROPOMI tropospheric $NO_2$ columns.

## 3 Results

In the following sections, we will present results of the simulations. We start with descriptions of the meteorological character-
istics of the four cases, specified for the time around TROPOMI overpass. Next, we will analyse the chemistry in the plumes
with a focus on the $NO_2$ lifetime and the $NO_x$:$NO_2$ ratio. Here, we will compare the simulated $CO_2$ plumes to the simulated
$NO_2$ plumes. Finally, we will compare the simulated $NO_2$ plumes to TROPOMI observations.

### 3.1 Meteorology

Simulations started at 0 UTC and lasted for 48 hours. Figure 3 shows the simulated wind speeds below 1000 m, averaged
over the model domains. Also indicated are the overpass times of TROPOMI. These wind speeds are strongly determined by
the boundary conditions that are provided by ERA5. Driven by the synoptic situation, winds in the lower boundary layer vary
considerably. We often observe a slow-down of the wind prior to TROPOMI overpass (vertical lines). This is related to the
growing convective boundary layer in the morning that propagates surface friction to higher altitudes. For LIP, winds are calm
prior to TROPOMI overpass on day 1, while the wind speed increases to more than 8 m s$^{-1}$ prior to TROPOMI overpass on
day 2. This is clearly reflected in the TROPOMI data in Fig. 2. Likewise, the lower TROPOMI columns for JAE on the second
day are caused by the larger wind speeds on day 2.

To analyse the situation further, Figure 4 shows profiles of wind speed, turbulent kinetic energy (TKE), and potential tem-
perature ($\theta$) sampled 30 minutes around the TROPOMI overpasses. TKE (m$^2$ s$^{-2}$) is calculated from the variances of the three
wind components:

$$TKE = \frac{1}{2}(\sigma_u^2 + \sigma_v^2 + \sigma_w^2). \tag{5}$$



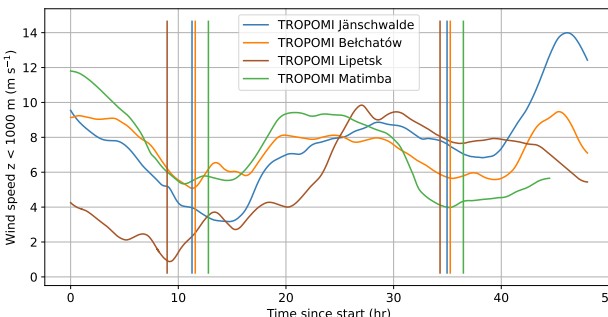

**Figure 3.** Time variation of the average wind speed at altitudes lower than 1000 m, horizontally averaged over the MicroHH model domains (Fig. 2). The vertical lines denote the UTC time of TROPOMI overpass on the two simulated days for each case.

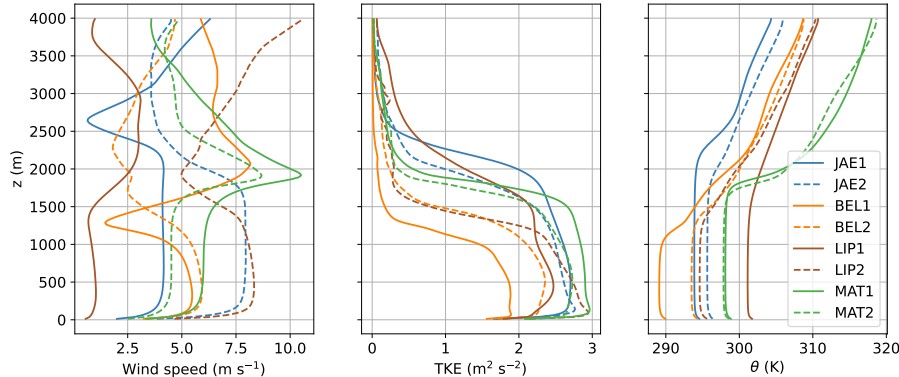

**Figure 4.** Domain-averaged profiles of wind speed (left), turbulent kinetic energy (TKE, middle), and potential temperature ($\theta$, right) averaged 30 minutes around TROPOMI overpass. Solid lines are for day 1 of the simulated cases, dashed lines for day 2.

215     In all cases, a well-mixed boundary layer is visible up to the inversions layer, with logarithmic profiles close to the surface. For instance, the boundary layer depth of the JAE1 case amounts to roughly 2500 m. We expect that, due to convective mixing, emissions from the stack will be distributed over the well-mixed boundary layer. Above the inversion layer winds either increase of decrease, and wind directions that change considerably with height (not shown). Despite the low wind speed, TKE is substantial in the LIP1 case, pointing to strong buoyancy. Turbulent mixing within the boundary layer is smallest in the case BEL1.

220     To derive a representative wind direction for plume dispersion, we determine this direction by weighting the wind-profile with the mean $NO_2$ profile. We subsequently rotate the MicroHH domain around the stack location using the plume direction angle, shown in Fig. 2, such that the plumes are aligned along the positive x-axis.



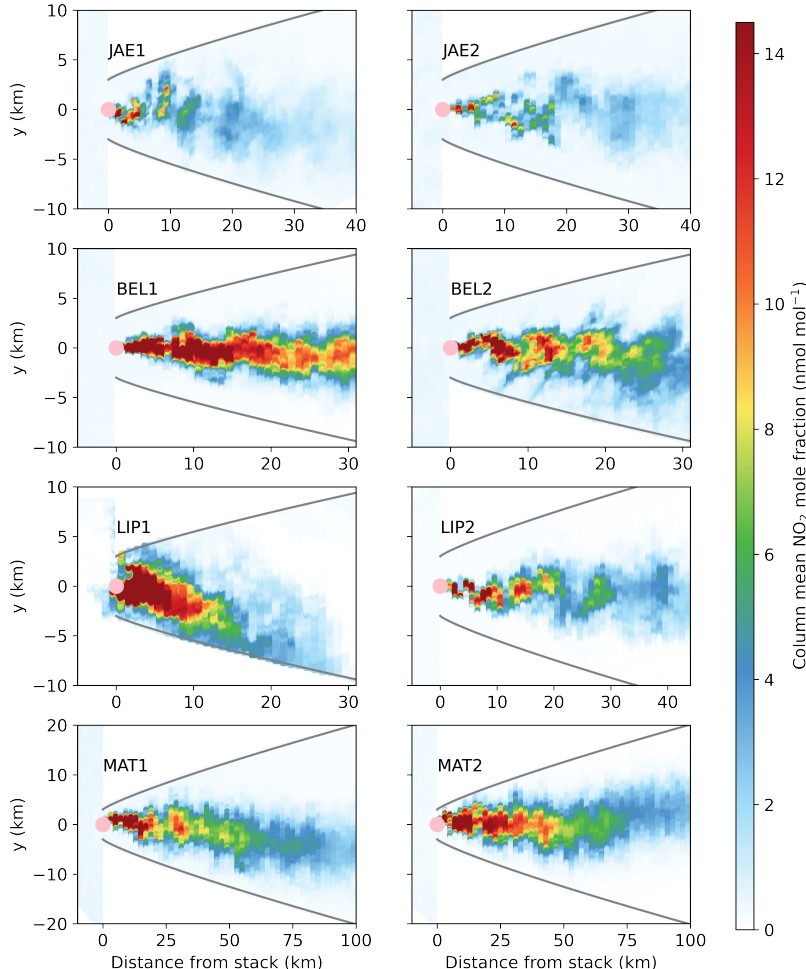

**Figure 5.** Simulated z-averaged NO$_2$ mole fraction of the simulated plumes at TROPOMI overpass time, aligned along the x-axis. The pink dots indicate the stack location. Mole fractions have been averaged between the surface and the height of the boundary layer. These boundary layer heights are respectively 2500 (JAE1), 2000 (JAE2), 1200 (BEL1), 1500 (BEL2), 1800 (LIP1), 1500 (LIP2), 1900 (MAT1), and 1850 m (MAT2) and are derived from Fig. 4. The black solid lines that encapsulate the plumes represent a Gaussian-type plume shape, and are given by the equation $y = \pm(3000 + 1.08 \times x^{0.84})$, with x and y in (m). Note that the x and y axes have different scales in the different panels.



## 3.2 Plume Chemistry

Figure 5 displays the simulated $NO_2$ mole fractions, averaged over the boundary layer at the times of TROPOMI overpass

25  (see Fig. 3). All plumes are aligned along the x-axis and share the same color scale. Except for the Lipetsk day 1 simulation (LIP1), which will be excluded in further analyses, the plumes stay within the Gaussian-type plume depicted by the black lines, which indicates that the winds are relatively stable in direction. The $NO_2$ abundance in the plume is mostly determined by the emission strength and the wind speed. However, as will be shown later, chemistry also plays and important role. The JAE1 and BEL2 plumes show more wavy lateral displacements compared to the other plumes, while the Matimba plumes reveal a slight

curvature, possibly due to effects of the Coriolis force (Potts et al., 2023).

To investigate the chemistry in the plume, cross-sections up- and down-wind of the stacks are analysed for the $NO_2$ and $NO_x$ lifetime, the mixing of $NO_2$ within the plume, and the abundance of OH. For all plume slabs downwind of the stack, averages are taken within the Gaussian-shape black lines in Figure 5 and bounded by the height of the boundary layer. Outside the plume upwind of the stack, the full domain up to the boundary layer height is considered. The lifetimes of $NO_x$ and $NO_2$ are

calculated by moles of $NO_x$ ($NO+NO_2$) or $NO_2$ (mole) divided by $NO_2$ loss in the reaction between $NO_2$ and OH (mole s$^{-1}$). The mean OH mole fraction represents the volume mean OH in these slabs. The mixing of $NO_2$ is quantified by calculating the intensity of segregation between OH and $NO_2$ in the slabs, which is defined as:

$$I_{s,NO_2,OH} = \frac{\overline{(NO_2 - \overline{NO_2})(OH - \overline{OH})}}{\overline{NO_2}\ \overline{OH}}. \tag{6}$$

Here, the bar represents a volume-average. $I_s$ thus represents the scaled covariance between two reactive compounds in a

volume (Danckwerts, 1952; Vilà-Guerau de Arellano et al., 1990; Galmarini et al., 1995; Krol et al., 2000; Ouwersloot et al., 2011). Generally, a negative value of $I_s$ signals a situation in which the concentrations of two reactants are negatively correlated, which implies that the chemical reaction between these species proceeds slower compared to a well-mixed situation. In contrast, a positive value of $I_s$ indicates that the reacting species are spatially correlated in a volume.

The $I_s$ concept thus quantifies the effect of assuming a well-mixed situation, e.g. in coarse-grid models. Specifically, if

$k_{NO_2,OH}$ represents the reaction rate between $NO_2$ and OH under well-mixed conditions, the modified reaction rate in a heterogeneously mixed air volume becomes:

$$k'_{NO_2,OH} = k_{NO_2,OH} \times (1 + I_{s,NO_2,OH}). \tag{7}$$

Figure 6 shows the calculated lifetimes of $NO_2$ (left panel) and $NO_x$ (right panel) in the simulated plumes. Right after emission, lifetimes show a clear spike. This is caused by the switch from a low/intermediate chemical $NO_x$ regime to a high-

$NO_x$ regime in the plume (McKeen et al., 1997; Vinken et al., 2011; Edwards et al., 2017; de Gouw et al., 2019). In this regime, $NO_2$ becomes the main sink for OH (reaction 12 in Table 3). Moreover, the concentration of $O_3$ drops to low values because of the reaction between $O_3$ and NO (reaction 6 in Table 3). Note that 90% of the $NO_x$ is emitted as NO.

Further downwind in the plumes, mixing of the plume with ambient air leads to a recovery of the $NO_x$ and $NO_2$ lifetimes. Even further downwind, lifetimes may become substantially shorter compared to ambient conditions. For instance, $NO_2$ life-

times converge to 1.5 hours for JAE1, JAE2, BEL2, and LIP2. This shorter $NO_x$ lifetime within the plume corresponds to




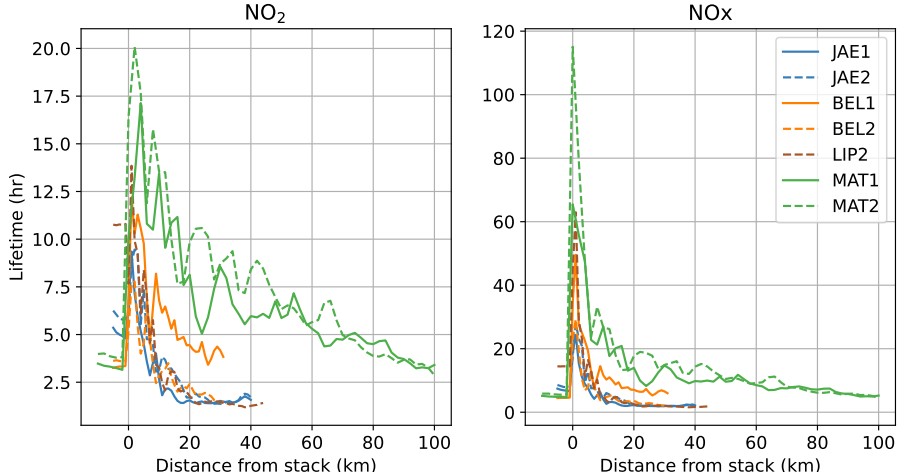

**Figure 6.** $NO_2$ (left) and $NO_x$ (right) lifetimes in the simulated plumes at the time of TROPOMI overpass, excluding LIP1. Lifetimes are calculated in volumes determined by 1 km slabs in the x-direction, the distance enclosed by the black lines in Fig. 5 in the y-direction, and the height of the boundary layer (see caption Fig. 5). Lifetimes are defined as moles of $NO_2$ or $NO_x$ (NO + $NO_2$) (mole) in this volume divided by chemical loss of $NO_2$ through the $NO_2$ – OH reaction (mole s$^{-1}$) in the same volume.

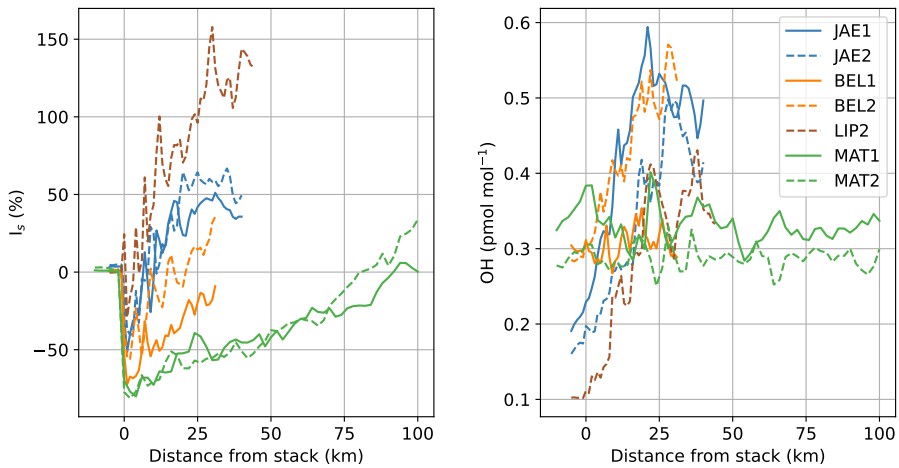

**Figure 7.** $I_{s,NO_2,OH}$ (in percent) and mean OH (in pmol mol$^{-1}$) as a function of distance from the stack location. Leftmost x-values smaller than zero represent background air. For definition of the volumes that were used for averaging, see Fig. 6. $I_s$ is defined in Eq. 6.

findings in de Gouw et al. (2019) that report faster removal of hydrocarbons in pollution plumes. The lifetime reduction depends strongly on the strength of mixing and the amount of $NO_x$ that is emitted at the stack. For Matimba, $NO_x$ emissions are very high (Table 5), which leads to a stronger perturbation of the plume chemistry and a slower recovery of the lifetimes. For



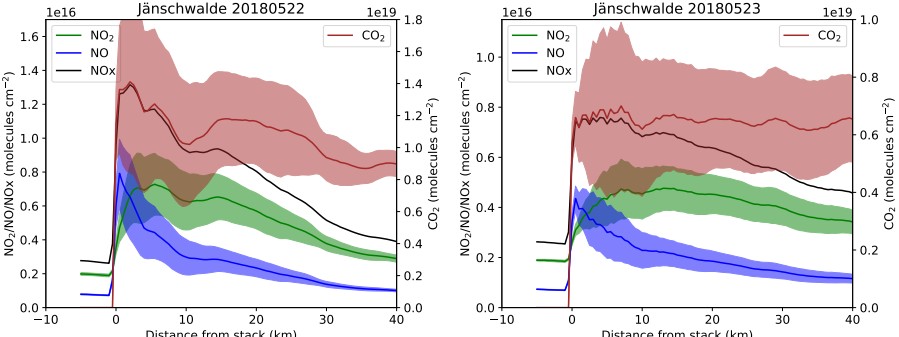

**Figure 8.** $CO_2$, NO, $NO_2$, and $NO_x$ columns up- and downwind of the Jänschwalde stack. Left and right panels refer to day 1 and day 2 of the simulation. Columns include the entire modelled column (0–4 km), and are averaged between y = −10 km and y = +10 km. Shaded areas (all species except $NO_x$) represent the $1\sigma$ temporal variability during one hour around TROPOMI overpass time. NO, $NO_2$, and $NO_x$ columns according to the left axes, $CO_2$ column according to the right axes. Note that the y-axes differ for both panels.

Bełchatów, the BEL1 plume stays intact longer compared to the BEL2 plume, driven by weaker mixing of the BEL1 plume
(Fig. 4). As a result, BEL1, MAT1, and MAT2 lifetimes remain longer than the background over the entire plume length.

Chemically, the behaviour of the lifetimes can be explained by the strong non-linear relation between the $NO_2$ abundance and its main sink OH. OH levels show a maximum at $NO_2$ mole fractions of 1–10 nmol mol$^{-1}$ due to recycling of OH (e.g. reaction 11) (Rohrer et al., 2014). Indeed, we observe this relation in our simulations, with low OH in the core plume (high-$NO_x$) and high OH concentrations at the plume edges, where due to intermediate $NO_x$ levels OH recycling is efficient.

To separate the effects of OH and mixing effects on the simulated lifetimes, Figure 7 shows slab-averaged $I_{s,NO_2,OH}$ (left) and OH (right) for the simulations. $I_s$ values vary strongly downwind of the stack. Starting from values close to zero outside the plume, values turn negative first, signaling anti-correlations between $NO_2$ and OH, in line with the high-$NO_x$ regime. For JAE1, JAE2, BEL2, and LIP2, $I_s$ values turn positive after $\approx$10 km. This implies a positive correlation between $NO_2$ and OH due to the strong recycling of OH in the chemical oxidation chain (Rohrer et al., 2014). In contrast, the BEL1, MAT1,
and MAT2 plumes show negative $I_s$ values, although values get gradually less negative at larger downwind distances and turn positive for the Matimba case at large distances from the stack. The split between intact and well-mixed $NO_2$ plumes also appears in mean OH mole fractions. In well-mixed plumes, mean OH in the plume is substantially enhanced further downwind of the stack, while OH stays roughly invariant in simulations BEL1, MAT1, and MAT2. For the latter plumes, the enhanced lifetimes (Fig. 6) are therefore mostly determined by $I_s$, while a combination of higher mean OH and $I_s$ is responsible for the
lifetime behaviour in the other plumes. Note that background lifetimes show large variations that are driven by differences in OH values outside the plume. For instance, the background $NO_2$ lifetime is $\approx$11 hours for LIP2, corresponding to an OH mole fraction of $\approx$0.1 pmol mole$^{-1}$. In contrast, values for MAT1 amount to $\approx$3.5 hours and $\approx$0.35 pmol mole$^{-1}$.

In a next step, we connect to methods that have been developed with the aim to quantify plume emissions from satellite data. For instance, in the cross-sectional flux (CSF) method described by Kuhlmann et al. (2020, 2021), the emission (in (mole s$^{-1}$))



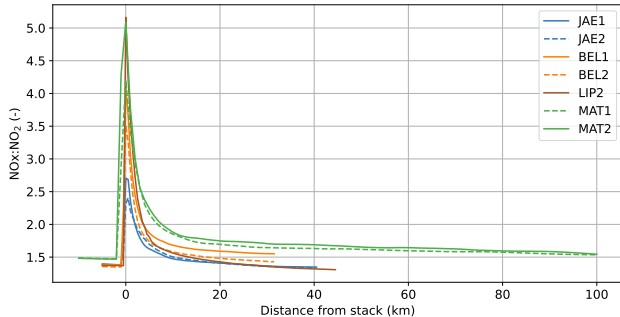

**Figure 9.** $NO_x$:$NO_2$ molar ratios for all simulations. Values are averaged over one hour around TROPOMI overpass, and for atmospheric slabs up- and downwind of the stack. These slabs extend to model top (4 km) and are averaged over y = [-10 km,+10 km] ([-20 km, +20 km] for MAT).

is derived from the integrals of the cross-section of the plume perpendicular to the wind direction (i.e. line density in (mole m$^{-1}$)) multiplied with an effective wind speed (in (m s$^{-1}$)). To investigate the validity of the underlying assumptions in these methods, Figure 8 shows column densities (0–4 km) of $CO_2$, NO, $NO_2$, and $NO_x$ (NO + $NO_2$), calculated as the mean in the cross-wind direction (y = [-10 km, 10 km], in molecules cm$^{-2}$) on the two days of the Jänschwalde simulation. Columns have been averaged during 1 hour around TROPOMI overpass, and the shaded areas denote $1\sigma$ variability (NO, $NO_2$, and $CO_2$). To allow comparison to $NO_x$, the right $CO_2$ axes have been scaled such that the $CO_2$ and $NO_x$ maxima match. As a result of the higher wind speed during the second day of the simulation, both y-axes in the right panel have smaller values (Fig. 4).

First, variability in the columns during this sampling hour is large, indicating a large role of turbulence. Due to turbulent eddies that are aligned with the wind direction, downwind transport of species from the stack is irregular, resulting in persistent patches of high concentration that move downwind (e.g. Fig. 5 and Cassiani et al. (2020)). Second, variability downwind of the stack decays on day 1, but remains sizable on day 2. Third, $CO_2$ columns in the plume are not constant on day 1, which shows that the gradual slowing down of the winds (see Fig. 3) has had a noticeable impact on the simulated columns. Assuming a mean wind speed of 5 m s$^{-1}$ in the atmospheric boundary layer (Fig. 3), the $NO_x$ and $CO_2$ as sampled 40 km downwind the stack was emitted more than two hours prior to TROPOMI overpass. Fourth, because of chemical removal of $NO_2$, $NO_x$ columns decay faster compared to $CO_2$ columns. Fig. 6 shows that this lifetime is not constant, but gets substantially shorter at larger distances from the stack, a feature that also shows up in Fig. 8. Finally, the $NO_x$:$NO_2$ ratio varies considerably along the plume. This is further corroborated in Figure 9, which shows the ratios for all simulations, except for LIP1.

For all plumes, $NO_x$:$NO_2$ ratios quickly rise from a background value of 1.3–1.5 to values of 3–5. Within the first 10 km, values decline to below 2, with further declines to background values further downwind of the stack. Interestingly, the downwind decay of $NO_x$:$NO_2$ ratios is slower for BEL1, MAT1, and MAT2, compared to the faster decaying plumes (JAE1, JAE2, BEL2, and LIP2). Thus, slow decaying plumes are characterised by persistent negative values of $I_s$, longer $NO_2$ and $NO_x$ lifetimes, and larger $NO_x$:$NO_2$ ratios. Both chemical factors (i.e. the size of the perturbation of the background chemistry) and mixing factors (i.e. mixing with ambient air) play a role is determining the plume behaviour.



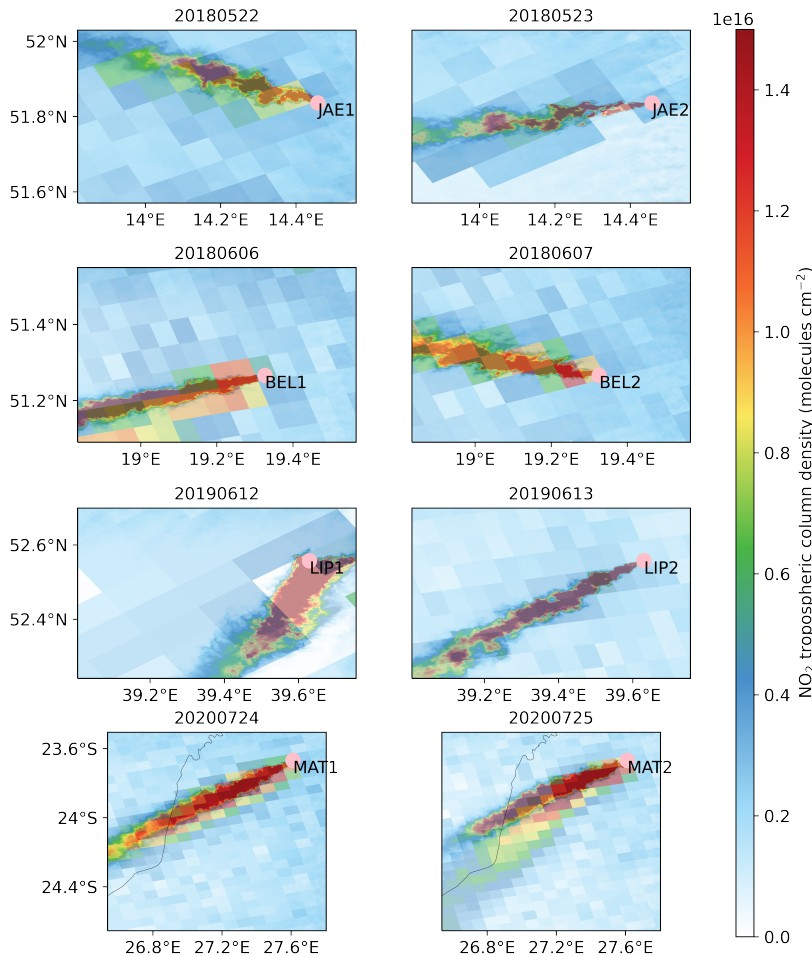

**Figure 10.** Same as Fig. 2, but now with simulated plumes overplotted on the same color scale. Note again that for the MAT case and the simulations, the columns are substantially outside the color range.

In the next section, we will compare the simulated $NO_2$ columns to TROPOMI, and evaluate the $NO_x$ emissions that were applied in the simulations.



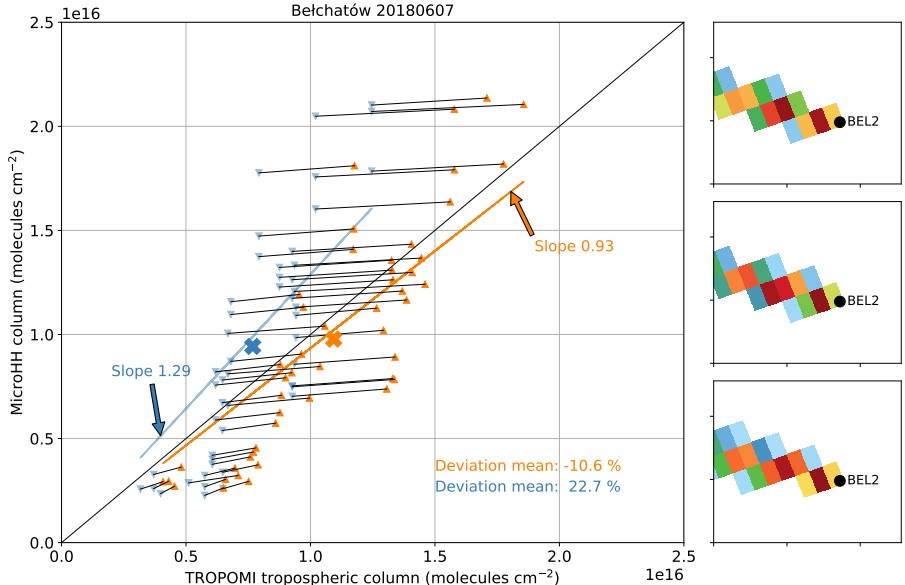

**Figure 11.** Comparison of simulated and TROPOMI tropospheric NO$_2$ columns for simulation BEL2. The three simulated plumes shown in the rightmost panels are 15 minutes apart around TROPOMI overpass time and are used to account for variations in the turbulent field. Domain and color bar as in Fig. 2, but in the range 0–2×10$^{16}$ molecules cm$^{-2}$. These fields are coarsened to TROPOMI resolution and filtered for enhanced NO$_2$ mixing ratios to identify the plume (see text). The blue dots and line show comparisons for TROPOMI data that have not been corrected for the AMF (Eq. 4). Moreover, simulated profiles have not been augmented with CAMS NO$_2$ in the free troposphere (i.e. 0–4 km). The orange dots show the corrected and augmented values, with corrected (orange) and uncorrected (blue) points connected by thin grey lines. Orange and blue lines and slope values represent fits that are forced through the origin. The orange and blue crosses represent the plume-mean columns and mean deviations ((model-tropomi)/tropomi in %) are given in the lower right corner.

## 3.3 Comparison to TROPOMI

As a first comparison between TROPOMI and the simulations, Figure 10 shows the model results at TROPOMI overpass time plotted on top of the TROPOMI NO$_2$ columns. Since the simulations are on 100 m resolution, more detail is visible in the simulations, and the NO$_2$ columns (even considering only z=0–4 km) are often outside the maximum color range. Generally, the simulated plumes align well with the observations. Only for BEL1 and MAT2, the simulated plume direction differs by roughly 10 degrees from the observed plume direction. For LIP1, the plume direction is ill-defined due to low wind speeds. Our simulations use meteorological boundary conditions from ERA5, and biases in ERA wind direction have been reported (Sandu et al., 2020). However, the way we impose the ERA5 boundary conditions using one time-dependent profile for the winds, also likely plays a role. As a consequence, when wind curvature is present in the ERA5 forcing fields, this curvature is currently not propagated to the MicroHH simulations. At larger scales, curvature due to the effect of the Coriolis force have been identified in TROPOMI images (Potts et al., 2023).



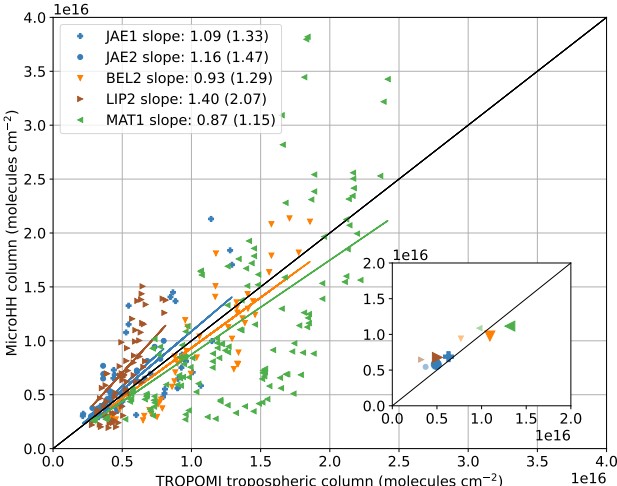

**Figure 12.** Similar to the main panel in Fig. 11, but now including comparisons for JAE1, JAE2, LIP2, and MAT1. Only corrected TROPOMI and MicroHH results are plotted, and the slope of the fit is given in the legend, with the slope for uncorrected data in parentheses. The inset shows the uncorrected (small transparent) and corrected (big symbols) plume means. Deviations after (before) correction are (in %) +9.3 (+31.5), +17.5 (+46.1), −10.6 (+22.7), +34.3 (+100.4), and −15.3 (+11.3) for JAE1, JAE2, BEL2, LIP2, and MAT1, respectively.

The next steps in the comparison between TROPOMI and simulated $NO_2$ plumes are a mapping of the simulated $NO_2$ fields to TROPOMI pixels, an extension of the simulated profiles to the tropopause, and an AMF correction of the TROPOMI columns using Eq. 4. We will present results for the cases JAE1, JAE2, BEL2, LIP2, and MAT1, based on the favorable match between model and TROPOMI. We focus on $NO_2$ enhancements above the background and filter for simulated mean column

mole fractions (0–4 km) smaller than 0.28, 0.25, 0.25, 0., and 0.28 nmol mol$^{-1}$ for JAE1, JAE2, BEL2, LIP2, and MAT1, respectively. These values differ slightly per case, because background $NO_x$ and winds vary in the simulations. To compare only the highly concentrated plume, we further discard TROPOMI tropospheric columns smaller than $2\times10^{15}$ molecules $NO_2$ cm$^{-2}$. To extend the simulated columns to the tropopause, CAMS $NO_2$ profiles are used. This extension adds a small and relatively constant amount of roughly $0.3\times10^{15}$ molecules $NO_2$ cm$^{-2}$. Since the amount of TROPOMI pixels that overlaps

with the simulated plumes is rather limited (e.g. only ≈17 for BEL2) we also use simulated fields 15 minutes before and after TROPOMI overpass.

Figure 11 shows the comparison between TROPOMI and the simulations for the BEL2 case and illustrates the effects of (i) adding free tropospheric columns from CAMS and (ii) the AMF correction of the TROPOMI columns with Eq. 4. The blue points denote the uncorrected TROPOMI $NO_2$ columns with the uncorrected simulations, while the orange points denote the

corrected values. Corrected and uncorrected values are connected with a thin grey line.

When comparing simulations to TROPOMI it should be realized that the simulations represent a highly turbulent field with large variability (e.g. Fig. 8) and that TROPOMI takes a low-resolution "snapshot" of this turbulent field. A clear one-to-one comparison is therefore not expected. Yet, the integrated or average columns should indicate whether the simulated $NO_2$




columns are systematically too high or too low. For this reason we calculated a linear fit (forced through the origin) and the
resulting slopes are given in Fig. 11. Moreover, we calculated the mean of the TROPOMI and MicroHH plumes, and results
are given by the blue and orange crosses in Fig. 11.

While the uncorrected slope would indicate a 29% overestimate of $NO_2$ columns in the simulation (i.e. too high $NO_x$ emissions), the corrected slope of 0.93 points to a slight underestimate. The change in slope is mostly caused by the AMF correction
of the TROPOMI tropospheric $NO_2$ columns. The correction factor is on average 1.40 (range 1.13–1.68) and corrections are
larger for enhanced $NO_2$ columns. This is caused by the fact that, in polluted conditions, a larger fraction of the $NO_2$ column
resides close to the surface in a high resolution model like MicroHH, compared to the coarse-scale TM5 model that is used
in the TROPOMI product (see Eq. 4). A similar conclusion can be drawn from the plume-mean columns: without correction, MicroHH overestimates the mean TROPOMI column by $\approx 23\%$, while after correction the mean TROPOMI column is
underestimated by $\approx 11\%$.

Figure 12 shows the results for all simulations. Like for BEL2, the calculated slopes reduce considerably when the AMF
correction is applied. Except for LIP2, slopes are within 20% of the 1:1 line, which would indicate that $NO_x$ emissions in the
Lipetsk simulation are too high. Results for JAE1 and JAE2 are rather consistent with slopes of $\approx 1.1$, while applied emissions
for Matimba might be slightly too low. The inset in Fig. 12 shows the plume-mean columns. Again, the AMF correction
generally improves the agreement between the simulations and TROPOMI, with signs of too high (low) emissions for the LIP2
(MAT1) simulation.

We note, however, that the spread of the individual pixels around the 1:1 line is considerable, with systematic model overestimates for high TROPOMI columns, and model underestimates for intermediate tropospheric $NO_2$ columns, specifically
for BEL2 and MAT1. This could point to either deficiencies in the model chemistry or transport, or to potential biases in the
TROPOMI columns. For instance, close to the stack, aerosols and clouds may have influenced the TROPOMI retrievals (Riess
et al., 2022; Geffen et al., 2022). Potential model deficiencies include a rather simple approach for plume rise. For both BEL1
and MAT1, winds aloft are stronger (see Fig. 4) and if plume rise would loft the emissions more, near-stack $NO_2$ columns
would decline while downwind columns would increase, potentially explaining some of the biases in Fig. 12. In general, however, a comparison between a turbulent plume and TROPOMI poses substantial challenges that need to be addressed in the
future. For instance, slight rotations or plume matching algorithms might allow for a more in depth evaluation of the emissions.
For now, we conclude that the emissions that are used in the simulations are likely within 20% of the true emissions, except for
Lipetsk. Deviations might be explained by the use of yearly average emissions in the simulations, while emissions may vary
considerably due to varying demand (Kuhlmann et al., 2021; Beirle et al., 2021). In the next section we will summarise and
discuss our main findings.

## 4  Discussion and Conclusions

In this section we discuss and draw conclusions by addressing the four questions that were posed in the introduction.



### 4.1 How does atmospheric chemistry affect the NO$_x$ plume?

Our simulations show that large NO$_x$ emissions in a background atmosphere lead to strong non-linear effects, in which the abundance of NO$_x$ strongly influences the NO$_x$ and NO$_2$ lifetimes, the NO$_x$:NO$_2$ ratio, and the duration over which a plume stays chemically intact. This latter effect is most clearly observed for the Matimba simulation in which the largest amount of
NO$_x$ is emitted (Table 5). The intensity of segregation between OH and NO$_2$ (I$_{s,NO_2,OH}$, Eq. 6) stays negative over more than 75 km downwind of the emission point, signalling plume regions that remain for a long time in the high-NO$_x$ chemical regime. Other plumes (JAE1, JAE2, BEL2, LIP2) show a high-NO$_x$ regime only in the first 5–10 km of the plume. At larger distance from the stack, a different chemical regime is present, with enhanced OH levels, positive I$_{s,NO_2,OH}$ values, and NO$_x$ and NO$_2$ lifetimes that are shorter than outside the plume (Fig. 6).

### 4.2 What is the impact of meteorology on plume dispersion?

Next to the NO$_x$ emission strength, 3D turbulence in the atmospheric boundary layer determines the dynamical and chemical behaviour of the simulated plumes. Although most simulated cases show well-mixed profiles of wind, TKE, and potential temperature in the boundary layer, some plumes are clearly more turbulent than others. For instance, simulation BEL1 shows less turbulent mixing compared to simulation BEL2 (Fig. 4). As a result, less ambient air is entrained in the plume, and the
plume stays longer intact, with persistent negative I$_s$ values and longer NO$_x$ and NO$_2$ lifetimes.

Another important meteorological factor is wind speed. We have identified that wind speed changes affect columns of an inert tracer like CO$_2$ substantially downwind of the plume (Fig. 8). For instance, the JAE1 simulation shows a substantial slowing down of the wind speed prior to TROPOMI overpass time (Fig. 3), which increases the CO$_2$ column by roughly 30% close to the stack location. To reduce errors in simplified methods that aim to quantify plume emissions from satellite
data (Kuhlmann et al., 2020, 2021), these methods should ideally account for these wind speed changes. On top of that, 3D turbulence in the boundary layer leads to large temporal variations in the simulated plumes. Simulated 1$\sigma$ variations (one hour averaging time) in CO$_2$ columns can easily reach 30%, with highest variability close to the stack location. This behaviour of turbulent plumes is well documented (Cassiani et al., 2020; Ražnjević et al., 2022a, b; Mu et al., 2023). Satellite images from polar orbiting platforms like TROPOMI and the upcoming CO2M mission (Sierk et al., 2019) take only a snapshot of
the turbulent plume, leading to uncertainties in simplified emission estimation methods (Kuhlmann et al., 2020). Large Eddy Simulations as presented in this paper help to identify the main factors that influence temporal variations in the plume and to design strategies to reduce errors in emission estimation methods.

### 4.3 How do the simulations compare to TROPOMI NO$_2$ observations?

Overall, we find a favourable comparison between the simulated plumes and TROPOMI (Fig. 10). The Lipetsk simulation
on day 1 had low wind speeds, which makes the comparison with TROPOMI difficult. Two other simulation days, BEL1 and MAT2, show that the simulated plumes are clock-wise rotated compared to the observations. Biases in simulated wind direction have been identified as a major source of uncertainty in other studies as well (Hakkarainen et al., 2023, 2019; Wu et al., 2023;





Brunner et al., 2023; Zheng et al., 2019; Lin et al., 2023). To allow a direct comparison between simulation and TROPOMI the implementation of a plume matching algorithm would be useful (e.g. Kuhlmann et al., 2021). Since our simulations are
nudged to a single time-dependent wind profile, wind rotation on the synoptic scale can currently not be resolved. Curvature observed in TROPOMI images has been attributed to effects of Coriolis forces (Potts et al., 2023). Efforts are ongoing to embed MicroHH within ERA5 & CAMS on larger domains with spatially varying forcing fields.

One of the largest challenges that has been identified in this study is that temporal variability in turbulent plumes is typically large, making a one-to-one comparison to satellite images difficult. TROPOMI samples in an afternoon orbit (13:30 local time
(LT)), while CO2M will have an overpass time of 11:30 LT (Sierk et al., 2019). Earlier overpass may avoid some of the strong turbulent plumes that we simulated here. However, other challenges remain.

In a simplified emission estimation procedure, we found that tropospheric TROPOMI columns should be enhanced by ≈40% due to different $NO_2$ amounts and profile shapes in the simulations (Eq. 4). Also here, however, there is no one-to-one match between a TROPOMI pixel and the simulation, which makes the applied correction uncertain. By averaging over
several simulation snapshots 15 minutes apart and by calculating the plume-mean enhancements, we tried to account for the modeled variability (Fig. 11). Comparisons showed that $NO_x$ emissions used in the simulations were likely correct within 20%, except for Lipetsk, for which the $NO_x$ emission in the model was ≈40% too high based on the comparison with TROPOMI. However, we also noticed systematic overestimates in the simulated columns close to the emission location, and systematic smaller columns at intermediate distances from the stack. Such biases may point to errors in our simplified chemistry and/or
TROPOMI retrievals and AMF correction. This latter might be due to the occurrence of plume-generated clouds and aerosol in the stack plume. One interesting finding that needs further exploration is the possible effect of plume rise on the vertical extend of the plume and its subsequent transport in the atmosphere. Simulated wind profiles (Fig. 3) show complex vertical structures. Better characterization and evaluation of the meteorological situation associated with point source emissions is therefore needed (e.g. Schalkwijk et al., 2015; van Stratum et al., 2023).

**4.4 What are the main factors that influence emission quantification from satellite observations?**

In the sections above, we identified the main factors that need to be accounted for in simulating plumes and comparison to satellite images.

First, wind speed and direction, and their variations in space and time drive how plumes are transported after emission. Associated to that, plume emission height and plume rise are important factors that have been identified before (Brunner et al.,
2019; Lin et al., 2023).

Second, we found that the $NO_x$ emission strength strongly determines the subsequent fate of the $NO_x$ plume. The large $NO_x$ emissions of the Matimba stack lead to a chemical perturbation that keeps the plume chemically intact for almost 100 km after emissions. Additionally, the turbulent mixing that mixes the plume with ambient air also plays a role, as shown for the Bełchatów simulation. We also identified plumes (JAE1, JAE2, BEL2, LIP2) that quickly move out of the high-$NO_x$
regime, and move into a regime with short $NO_x$ and $NO_2$ lifetimes. Accounting for proper $NO_x$ lifetimes and $NO_x$:$NO_2$ ratios is important to use $NO_2$ as an additional tracer to constrain $CO_2$ stack emissions.



Third, the AMF correction of TROPOMI data is an important factor. However, variability in the simulated $NO_2$ distribution makes a one-to-one comparison to TROPOMI images, and hence a proper AMF correction, difficult. New data-assimilation techniques to better constrain 3D turbulence are being developed (Chandramouli et al., 2020; Bauweraerts and Meyers, 2021),
but require time-resolved observations of 3D turbulence. One option to explore is the use of data from recently reported time-resolved imaging spectroscopy (Mu et al., 2023). Our chemistry-enabled MicroHH simulations in combination with these detailed observations may improve methods to quantify emissions from large point sources.

Our study also has some shortcomings. First, our chemistry is simplified, and does not account for possible impacts of heterogeneous reactions on aerosol surfaces. In highly concentrated plumes, these processes may be important (Kim et al., 2017). Sec-
ond, we performed our simulations according to the CoCO2 simulation protocols (https://coco2-project.eu/sites/default/files/2021-07/CoCO2- D4.1-V1-0.pdf) that only crudely account for plume rise. In the future, we could add heat and moisture stack emissions to the simulations to account explicitly for plume rise. Third, next to applying inflow due to CAMS boundary conditions, we only accounted for emissions from the stack, ignoring possible surface emissions. As a result, concentrations fields might be less realistic, specifically in the downwind domain outside the plume. Fourth, the use of 100 m resolution LES at night
is insufficient to resolve small scale turbulence in the nocturnal boundary layer. As a result, transport will be driven by the sub-grid model, leading to overestimation of mixing. However, the impact on the development of a convective boundary layer on the next day was shown to be limited (van Stratum and Stevens, 2015). Finally, our boundary conditions consist of single time-dependent columns. Thus, the simulations cannot account for commonly-observed rotation in the wind fields. Future developments of the MicroHH code will account for these shortcomings.

In conclusion, we presented LES simulations of $NO_2$ plumes from four large emitters world-wide. To this end, we implemented a simple chemistry scheme in the MicroHH model. Simulations showed generally good agreement with TROPOMI images, and the need to account for the strongly non-linear $NO_x$ chemistry in concentrated plumes. LES simulations with chemistry are useful to test less involved algorithms to derive emissions from large point sources. As a start, the use of a fixed $NO_2$ lifetime and $NO_x:NO_2$ ratio can be replaced by values derived from our high-resolution plume simulations.

*Code and data availability.* The MicroHH code used for the calculations is available from GitHub (https://github.com/microhh/microhh, branch develop_kpp) and is also deposited at Zenodo, together with a python Notebook, model input, and model output, that was used to produce the figures (https://doi.org/10.5281/zenodo.10053684).

*Author contributions.* MK performed the calculations and wrote the manuscript. BvS coordinated the chemistry implementation within MicroHH. IA helped with the AMF correction of TROPOMI, and KFB helped with the interpretation of TROPOMI. All co-authors commented
on the draft manuscript.



*Competing interests.* We declare that no competing interests are present

*Acknowledgements.* Work described in this paper was performed in the CoCO2 project that received funding from the European Union's
Horizon 2020 research and innovation programme under Grant Agreement No. 958927. This publication is part of the project NSOKNW.2019.002
of the research programme PIPP which is (partly) financed by the Dutch Research Council (NWO). Chiel van Heerwaarden is acknowledged
for assisting with the numerical implication of the chemistry within MicroHH. Model runs were performed on the Dutch National Super-
computer Snellius. We would like to acknowledge SurfSARA Computing and Networking Services for their support. CoCO2 partners are
acknowledged for setting up the modelling protocols.



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
