# Peer review of "Evaluating $NO_x$ stack plume emissions using a high-resolution atmospheric chemistry model and satellite-derived $NO_2$ columns"

_EGUsphere, 2023_

## Referee Comment (RC1)

Review of "Estimating NOx emissions of stack plumes using a high-resolution atmospheric chemistry model and satellite-derived NO$_2$ columns" by Krol et al.

This manuscript proposes a novel method for estimating NOx emissions from point sources by using a Large-Eddy Simulation (LES) model in conjunction with satellite-derived NO$_2$ column data. The authors conducted simulations of plumes from four large power plants with NOx emissions incorporating related chemistry, and compared the model results with TROPOMI tropospheric NO$_2$ columns. The model columns agreed reasonably with the satellite columns that adopted the improved air mass factors. Considering uncertainties both in the model simulations and satellite retrievals, the level of agreement between the two are quite encouraging. The manuscript introduces several innovative approaches to address the research question and thoroughly discussed the results including uncertainties and limitations of this approach. I would be happy to recommend the publication of this manuscript to ACP after minor revisions. I summarized the points to be revised.

1) The abstract of this manuscript is quite distracting. All CO$_2$ related remarks should go to the later part of introduction or discussion section. The main topic in this manuscript is NOx emissions from point sources related to LES model simulations and TROPOMI tropospheric NO$_2$ column observations. Furthermore, the sentence "Moreover, results indicate that common assumptions about the NO$_2$ lifetime (~4 hours) and NO$_x$:NO$_2$ ratios (~1.3) in simplified methods that estimate emissions from NO$_2$ satellite data (e.g. Beirle et al., 2019) need revision" needs to be revised. This targets only specific studies and does not give broad implications and directions.

2) This manuscript deals with the classical nonlinear relationship between NOx and OH. The authors several times referred to Rohrer et al. (2014) for recycling of OH. Rohrer et al. (2014) is mainly concerning about a new recycling process generating OH under very low NOx and high biogenic VOC condition. I don't think that this manuscript is closely related to Rohrer et al (2014). There would be better references for this. Meanwhile, NOx lifetime estimations and related discussions based on satellite observations can be found in Valin et al (2013) and Laughner and Cohen (2019) and references therein.

L. C. Valin et al., Geophys. Res. Lett. 40, 1856-1860 (2013).

Laughner and Cohen, Science 366, 723-727 (2019).

3)  It would be beneficial to include the plot for BEL2 from Figure 2 alongside the three plots of the LES model $NO_2$ columns (projected to the TROPOMI pixels) in Figure 11. This addition would facilitate one-to-one comparison.

4) It is not clear that the emission the model used is based on the bottom-up emission inventories and the one measured at each stack. If we consider TROPOMI tropospheric $NO_2$ columns as a ground truth value, the agreement between the model and satellite gives a confidence in the bottom-up emission from this power plant, probably as shown in this study. While the TROPOMI data appear promising, additional validation and calibration would be necessary, particularly for observations near power plants. If the authors utilized the observed NOx emission from the power plants for their simulations, the agreement between the model and satellite data provides insight into the accuracy of the TROPOMI columns. Thus, the story changes, depending on the nature of emission in the model.

5) The analysis of Is and $NO_2$ lifetimes for the different power plants is valuable. However, it would be better if Is and $NO_2$ lifetime are also calculated for the pixels of TROPOMI or larger source box (like 100 km x 100 km) for interpretation of real-world problems. Beirle et al (2019) noted the specific condition for which the assumption of NOx/NO2 = ~ 1.3 is valid. It is needed to interpret the results in this study in line with Beirle et al (2019) or similar studies and other research that adopted larger source boxes.

6) The recommendations for future calculations of NOx emissions from stack plumes remain unclear in this study. Should the LES method be applied for all power plants worldwide? Can TROPOMI tropospheric $NO_2$ columns sufficiently provide NOx emissions estimations from these sources?

---

## Author Comment (AC2)

**Authors' Response to Reviews of**

**Estimating $NO_x$ emissions of stack plumes using a high-resolution atmospheric chemistry model and satellite-derived $NO_2$ columns**

Maarten Krol, Bart van Stratum, Isidora Anglou, Klaas Folkert Boersma
*Egusphere, 2024,* `https://doi.org/10.5194/egusphere-2023-2519`
* * *
**RC:** *Reviewers' Comment*,     AR: Authors' Response,     ☐ Manuscript Text

**1. Reviewer #3**

We thank the reviewer for the positive evaluation of our manuscript. Concerning the minor revisions required, we answer point by point below.

**1.1. Main comments**

**RC:** *Title: The title of the manuscript centered on the estimate of NOx emissions, but the current manuscript seems to focus less on the inversion/optimization of NOx emissions, and more on the forward simulations of NO2 columns and how transport, chemistry, and adaptation to TROPOMI affect the simulations. I may suggest the authors revise the title to better highlight the real novelty of their study.*

**AR:** We agree with this statement and changed "Estimating" in the title to "Evaluating", since this better reflects what we did:

> Evaluating $NO_x$ stack plume emissions using a high-resolution atmospheric chemistry model and satellite-derived $NO_2$ columns

**RC:** *Methodology: The definition of plume background area and the choice of prior emissions seem to be unclear. Did the authors use "correct" emissions reported from the CoCO2 project? If not, since emission, mixing, and chemistry can all be uncertain, it would be hard to evaluate the model performance (especially given the end goal of estimating emissions). Also, the "background" or background air has been mentioned but not properly defined. Was there a fixed threshold or a fixed distance from the stack? The background definition may affect the statements about lifetime diagnoses (page 18).*

**AR:** Concerning the emissions, see our answer to Reviewer #2, point 4. We conclude that we used the best available information as emission estimates. For LIP, these numbers are more uncertain. Concerning the background, we supply boundary conditions from ERA5 (meteorology) and CAMS (chemistry), like outlined in lines 156–161 of the manuscript. In response to a reviewer question on another paper (Meier et al., 2024), we provided this information, which can be found under `https://doi.org/10.5194/egusphere-2024-159-AC1`, Supplementary Figure S2.Thus, we used boundary conditions from reanalysis products, in which our simulations are embedded. We did not test the sensitivity of our results to these boundary conditions.

Concerning the location of the stack, we selected the domain such that we kept some distance from the boundary to allow the (circular) turbulent field to interact with the prescribed tracer profiles at the boundary. Furthermore, we selected the domain such that – with the prevailing wind condition – the plume had a substantial length within the domain.

Since we feel that the information in the paper is already adequate to address the issues raised, we did not modify the manuscript.

**RC:** *Results: I understand the authors' great effort in adding chemical reactions in LES for several simulations, but the explicit estimation of uncertainties associated with transport/mixing and chemistry seem to be missing from the manuscript, especially when the authors also acknowledged large uncertainties in the chemical model parameterization and atmospheric transport (L43 – L45). Again, if the final goal is to estimate/optimize emissions as suggested by the title, it would be important to acknowledge uncertainties in every model component (using either a simple CSF method or more complex chemical inversions). For example, Wu et al. (2023) (as mentioned on L44) tested the WRF-Chem-based chemical parameterization by estimating the chemical uncertainties with NOx lifetime and NO2-to-NOx ratio and by evaluating against TROPOMI using EPA-reported power plant emissions.*

**AR:** We fully understand this comment. We evaluated the uncertainties associated with chemistry and mixing in more detail than reported in the manuscript. However, we wanted to avoid the paper becoming too technical and only briefly reported these results. We will now include Supplementary information to present these results. Below and in the Supplement, we present figures that show evaluations of the chemistry using a box model (with different emission scenarios compared to Figure 1 in the original paper). We simulated the chemistry of the full IFS chemistry scheme (IFS) and our simplified version presented in the paper (MHH). The box model considers emissions of NO and hydrocarbons (RH = $C_3H_6$) at the surface of a 1000 m well-mixed atmospheric box. Diurnal variations in light conditions are accounted for and simulations were run for two days. Figures 1– 9 show the results with varying emission strengths, covering both ozone depletion and ozone production regimes. These regimes are expected in our plume simulations. Like mentioned in the original manuscript, results are almost identical when no emissions of hydrocarbons are considered (ERH = 0). Results deteriorate when emissions of hydrocarbons are considered, but remain acceptable (errors smaller than 10% for $NO_2$) specifically for $NO_2$ (sampled by TROPOMI) and when only the first day is considered. Here we note that air masses stay relatively short in our domain and that air masses forced by the CAMS boundary conditions that enter the domain.

We also evaluated the effects of resolution on mixing and chemistry in the plume. These results have been obtained with a controlled LES setup with a prescribed west–east wind (10 m s$^{-1}$ geostrophic wind) and prescribed boundary conditions for temperature and chemical species. The simulation domain size was $[\Delta x, \Delta y, \Delta z]$ = [9600 m, 4800 m, 3200 m], i.e. much smaller compared to the simulation domain in the main paper. These simulations have been performed on different resolutions (200, 100, 50, 25 m) to test the effects of resolution on the main findings. The simulations were forced with a surface heat flux of 0.1 K m s$^{-1}$ and emissions from a 200-m high stack, located 1600 m from the west border. The source magnitude was 50% of the MAT emissions. The simulations were performed for 4 hours (starting at 6 am) and the results were averaged over the last hour of the simulation.

First, Figure 10 shows the $NO_2$ column averaged over the last hour of the simulation. Although the plumes appear similar at different resolutions, the conversion of NO to $NO_2$ proceeds slower at a finer resolution. This is corroborated in Figures 11, 12, and 13, which show the y-mean columns of, respectively, $NO_2$, NO, and $O_3$. Due to emission, sharp concentration gradients form at the stack location, because $O_3$ levels are titrated by the high supply of NO. Note that 95% of the $NO_x$ emission is emitted as NO. These sharp gradients are expected to be resolved better in the high-resolution simulation. Results seem to converge at a resolution of 25 m. At a resolution of 100 m, as used in the simulations presented in the paper, a substantial instant dilution error is still present. For instance, at a distance of 6000 m from the stack, y-mean $NO_2$ columns are $\approx$20% larger at 100 m resolution, compared to the 25 m resolution simulation. These errors are hard to avoid if you want to address a larger domain of 50–100 km because the calculation time quickly becomes

[Figure]

Figure 1: Box model comparison (h = 1000 m) of the IFS scheme with the condensed MicroHH (MHH) scheme. Results of a two day simulation are shown, starting at 8 AM. Emission strengths of NO and $C_3H_6$ are given in the figure title. Time series are plotted for NO, $NO_2$, $O_3$, OH, HCHO, and RH ($C_3H_6$).

prohibitive. The 100 m resolution used in the simulations presented in the paper is therefore a compromise, and this issue clearly warrants additional investigation. Concerning the ratio $NO_x$ to $NO_2$ discussed in the paper, the effect of resolution is also important. Figure 14 shows the calculated ratio downwind of the stack. Although the $NO_x$ to $NO_2$ ratio shows a clear spike at the emission point, this spike is smaller at 100 m and 200 m resolutions, and the ratio returns faster to background values.

In conclusion, at higher spatial model resolution, the conversion of NO to $NO_2$ is expected to proceed slower, and the ratios of $NO_x$ to $NO_2$ are expected to be substantially larger at distances up to 10 km downwind of the stack. These effects likely depend on the emission strength, the background ozone concentration, the wind speed, and the stability of the boundary layer, and warrant further investigation.

We will add a Supplement to the revised manuscript to present these results, and adjust the main text to convey this message more clearly.

**1.2. Minor comments**

**RC:** *Eq. 2: What are some other processes? Emission and deposition?*

AR: As written, MicroHH collects tendencies for all processes and uses the RK3 solver to progress in time. Processes include advection and sources. Deposition is treated as part of the chemistry. We modified line 121: "The calculation of the chemistry tendencies is evaluated after the calculation of all other tendencies, including emissions." to:

[Figure]

Figure 2: As Figure 1.

> The calculation of the chemistry tendencies is evaluated after the calculation of all other tendencies, like advection and emission tendencies.

**RC:** *There seemed to be some considerations of the injection height (L78) and the use of a plume rise calculation (Table 5). But whether/how both injection height and the plume-rise were implemented in the model is not super clear to me, despite the acknowledgment and discussions in Sect. 4.4. More clarifications would help the readers, as injection height and plume height are important for point sources with intensive emission rates.*

**AR:** Since not all participating models in the underlying intercomparison study could calculate plume rise in a uniform way, we follow the CoCO2 intercomparison protocols (e.g. line 135, `https://coco2-project.eu/sites/default/files/2021-07/CoCO2-D4.1-V1-0.pdf`). To outline the general approach that led to the profiles in Table 5, we will add to the revised manuscript (above table 5):

> Concerning emission height, we prescribed fixed emission profiles that account for plume rise. For JAE and BEL, these were calculated using the empirical equations recommended by the Association of German Engineers, which are based on the original work of Briggs (1984). Typical stack parameters were obtained from Pregger and Friedrich (2009), considering typical power plant capacities and fuel types, and from site descriptions. For LIP and MAT, the emission heights recommended by the CAMS emission dataset (Kuenen et al., 2022) for Industry and Public Power sectors were used.

**RC:** *Future application: Just out of general curiosity, how is the possibility of applying this approach to urban areas?*

**AR:** Yes, we are currently working on the implementation of our method to urban areas. We will add a sentence near the end of the revised manuscript to point this out:

[Figure]

Figure 3: As Figure 1.

Moreover, we are extending our method to more complicated emission hot spots, like cities.

**References**

Briggs, G.: Plume Rise and Buoyance Effects, pp. 327–366, U.S. Department of Energy, 1984.

Kuenen, J., Dellaert, S., Visschedijk, A., Jalkanen, J.-P., Super, I., and Gon, H. D. V. D.: CAMS-REG-v4: a state-of-the-art high-resolution European emission inventory for air quality modelling, Earth Syst. Sci. Data, 14, 491–515, , 2022.

Meier, S., Koene, E. F. M., Krol, M., Brunner, D., Damm, A., and Kuhlmann, G.: A light-weight NO2 to NOx conversion model for quantifying NOx emissions of point sources from NO2 satellite observations, EGUsphere [preprint], -, , 2024.

Pregger, T. and Friedrich, R.: Effective pollutant emission heights for atmospheric transport modelling based on real-world information, Environmental Pollution, 157, 552–560, , 2009.

ERH = 0.3, ENO = 0.1 ppb m/s

[Figure]

Figure 4: As Figure 1.

ERH = 0.3, ENO = 0.3 ppb m/s

[Figure]

Figure 5: As Figure 1.This figure corresponds to Figure 1 in the main paper.

ERH =    0.3, ENO =    0.5 ppb m/s

[Figure]

Figure 6: As Figure 1.

ERH =    0.5, ENO =    0.1 ppb m/s

[Figure]

Figure 7: As Figure 1.

ERH =    0.5, ENO =    0.3 ppb m/s

[Figure]

Figure 8: As Figure 1.

ERH =    0.5, ENO =    0.5 ppb m/s

[Figure]

Figure 9: As Figure 1.

[Figure]

Figure 10: Total column $NO_2$ averaged over the last hour of a 4 hour simulation on a small domain. Emission with a source strength of half the MAT emissions were added at [x,y,z] = [1600 m, 2400 m, 200 m]. Simulations were performed on resolutions of 25 m (top panel), 50 m, 100 m, and 200 m (bottom panel).

[Figure]

Figure 11: NO$_2$ y-mean columns from Fig. 10. The dotted line indicates the stack location.

[Figure]

Figure 12: NO y-mean columns from Fig. 10. The dotted line indicates the stack location.

[Figure]

Figure 13: $O_3$ y-mean columns from Fig. 10. The dotted line indicates the stack location.

[Figure]

Figure 14: Y-mean $NO_x$ over $NO_2$ ratio. The dotted line indicates the stack location.

---

## Author Comment (AC3)

**Authors' Response to Reviews of**

**Estimating $NO_x$ emissions of stack plumes using a high-resolution atmospheric chemistry model and satellite-derived $NO_2$ columns**

Maarten Krol, Bart van Stratum, Isidora Anglou, Klaas Folkert Boersma
*Egusphere, 2024,* `https://doi.org/10.5194/egusphere-2023-2519`
* * *
RC: *Reviewers' Comment*,  AR: Authors' Response,  ☐ Manuscript Text

**1. Reviewer #2**

We thank the reviewer for the positive evaluation of our manuscript. Concerning the minor revisions required, we answer point by point below.

**RC:** *1) The abstract of this manuscript is quite distracting. All CO2 related remarks should go to the later part of introduction or discussion section. The main topic in this manuscript is NOx emissions from point sources related to LES model simulations and TROPOMI tropospheric NO2 column observations. Furthermore, the sentence "Moreover, results indicate that common assumptions about the NO2 lifetime ( 4 hours) and NOx:NO2 ratios ( 1.3) in simplified methods that estimate emissions from NO2 satellite data (e.g. Beirle et al., 2019) need revision" needs to be revised. This targets only specific studies and does not give broad implications and directions.*

**AR:** The reviewer makes a valid point here. Indeed, the final manuscript is about chemistry-enabled LES, focusing on comparisons to TROPOMI $NO_2$ satellite data. We will rewrite the abstract and introduction accordingly.

We do not fully grasp what is meant by "This targets only specific studies and does not give broad implications and directions". A recent study (Meier et al., 2024) based on the output of our LES runs develops a general algorithm to account for the $NO_x$:$NO_2$ ratio in the analysis of the TROPOMI data. In addition, the role of the $NO_2$ lifetime seems to be of general importance in the interpretation of $NO_2$ satellite data, as has been shown in many papers. However, we will omit the reference from Beirle et al. 2019 and simply refer to 'the common assumptions'.

**RC:** *2) This manuscript deals with the classical nonlinear relationship between NOx and OH. The authors several times referred to Rohrer et al. (2014) for recycling of OH. Rohrer et al. (2014) is mainly concerning about a new recycling process generating OH under very low NOx and high biogenic VOC condition. I don't think that this manuscript is closely related to Rohrer et al (2014). There would be better references for this. Meanwhile, NOx lifetime estimations and related discussions based on satellite observations can be found in Valin et al (2013) and Laughner and Cohen (2019) and references therein.*

**AR:** We replaced the Rohrer et al. reference by other references to give proper credit to the historical developments. Specifically, we will refer to: Ehhalt and Rohrer (2000); Lelieveld et al. (2002); Valin et al. (2013).

**RC:** *3) It would be beneficial to include the plot for BEL2 from Figure 2 alongside the three plots of the LES model NO2 columns (projected to the TROPOMI pixels) in Figure 11. This addition would facilitate one-to-one comparison.*

**AR:** Thanks for the suggestion. Below updated figure 11.

[Figure]

Figure 1: Comparison of simulated and TROPOMI tropospheric $NO_2$ columns for simulation BEL2. The upper right panel repeats the TROPOMI data from Fig. 2. The three panels below are simulation snapshots 15 minutes apart around TROPOMI overpass time and are used to account for variations in the turbulent field. Domain and color bar as in Fig. 2, but in the range $0–2\times10^{16}$ molecules $cm^{-2}$. These fields are coarsened to TROPOMI resolution and filtered for enhanced $NO_2$ mixing ratios to identify the plume (see text). The blue dots and line show comparisons for TROPOMI data that have not been corrected for the AMF (Eq. 4). Moreover, simulated profiles have not been augmented with CAMS $NO_2$ in the free troposphere (i.e. 0–4 km). The orange dots show the corrected and augmented values, with corrected (orange) and uncorrected (blue) points connected by thin grey lines. Orange and blue lines and slope values represent fits that are forced through the origin. The orange and blue crosses represent the plume-mean columns and mean deviations ((model-tropomi)/tropomi in %) are given in the lower right corner.

RC: **4) It is not clear that the emission the model used is based on the bottom-up emission inventories and the one measured at each stack. If we consider TROPOMI tropospheric NO2 columns as a ground truth value, the agreement between the model and satellite gives a confidence in the bottom-up emission from this power plant, probably as shown in this study. While the TROPOMI data appear promising, additional validation and calibration would be necessary, particularly for observations near power plants. If the authors utilized the observed NOx emission from the power plants for their simulations, the agreement between the model and satellite data provides insight into the accuracy of the TROPOMI columns. Thus, the story changes, depending on the nature of emission in the model.**

AR: As written in the manuscript, the simulations were performed based on the protocols that were written at the start of the CoCO2 project (`https://coco2-project.eu/sites/default/files/2021-07/CoCO2-D4.1-V1-0.pdf`). Our trust in the emissions used in the simulation depends on the case. As also outlined in Meier et al. (2024), estimated emissions of the European power plants Jänschwalde and Bełchatów are based on yearly data from the European Pollutant Release and Transfer Register (E-PRTR). For these simulations, we used the bottom-up reported emissions to compare to TROPOMI $NO_2$ columns to detect

possible discrepancies. We obtained data as annual $NO_x$ emissions from the Jänschwalde power plant for the year 2018. For the Bełchatów power plant, the data are only available up to 2017, and emissions for 2017 were used.

For the Matimba case, we use the emissions averaged over the year 2018, based on the monthly reports provided by the responsible company Eskom (https://www.eskom.co.za). Earlier studies (Hakkarainen et al., 2021) showed that TROPOMI-based annual $NO_x$ emissions are slightly lower than the annual value reported by the company. However, the difference remained within the errors in the estimation method.

However, for the Lipetsk metallurgical plant, we had to make a rough estimate because no accurate data on emissions are available. We therefore estimated the emission on the basis of an annual report of the operating company NLMK. However, from the report, it is unclear whether the reported emissions are exclusively from the metallurgical plant in Lipetsk only and whether emissions from the captive power plants at the Lipetsk site are included in the reported emissions. Thus for the Lipetsk case, TROPOMI tropospheric $NO_2$ columns are used as a ground truth value. Reinforcing the accuracy of TROPOMI and the comparison procedure, we find the poorest agreement for the Lipetsk case.

To clarify this issue in the manuscript, we added the following lines to the document

> The emissions are based on the CoCO2 intercomparison protocol: `https://coco2-project.eu/sites/default/files/2021-07/CoCO2-D4.1-V1-0.pdf`. For JAE and BEL,these emissions are based on reported yearly total values in the European Pollutant Release and Transfer Register (E-PRTR). For MAT, we used the average emissions for the year 2018, based on the monthly reports provided by the responsible company Eskom (`https://www.eskom.co.za`). For LIP, emissions are obtained from the 2019 annual report of the NLMK group (`https://nlmk.com/en/ir/reporting-center/annual-reports/`). We have smaller confidence in these LIP emissions, because it remains unclear whether the reported emissions can be fully ascribed to the Lipetsk facility.

**RC:** *5) The analysis of Is and NO2 lifetimes for the different power plants is valuable. However, it would be better if Is and NO2 lifetime are also calculated for the pixels of TROPOMI or larger source box (like 100 km x 100 km) for interpretation of real-world problems. Beirle et al (2019) noted the specific condition for which the assumption of NOx/NO2 = 1.3 is valid. It is needed to interpret the results in this study in line with Beirle et al (2019) or similar studies and other research that adopted larger source boxes.*

**AR:** We tend to disagree with this argument. For the problem of mixing chemicals in a plume ($I_s$) and the $NO_2$ lifetime, we considered it useful to analyse what happens as a function of distance from the stack at high spatial resolution, and this is what we did in the paper. The question of how the overall effect is observed by a satellite instrument, such as TROPOMI, is another issue. We consider it, however, instructive to show the implications on TROPOMI resolution. We will show these results in a Supplement to the manuscript. Figures 2, 3, 4, and 5 show results for $I_s$, the lifetime of $NO_2$, the lifetime of $NO_x$, and the $NO_x$ to $NO_2$ ratio, respectively. Note that for the MAT case, the lifetime of $NO_2$ in the background gets very long, due to depletion of $NO_x$ by oxidation in the extensive domain. As mentioned in the manuscript, we focused on emission of a single stack, and added no surface emissions of e.g. traffic. Note also that these figures confirm the finding in the paper that the BEL1, MAT1, and MAT2 plumes remain chemically intact for larger downwind distances.

**RC:** *6)The recommendations for future calculations of NOx emissions from stack plumes remain unclear in this study. Should the LES method be applied for all power plants worldwide? Can TROPOMI tropospheric NO2 columns sufficiently provide NOx emissions estimations from these sources?*

AR:   We believe that our paper, for the first time, applies a very-high resolution model (much finer than TROPOMI resolution) to assess the effects of non-linear $NO_x$ chemistry and mixing on the evolution of chemical plumes. Results of this paper are currently used to derive improved parameterisations for simplified methods, and results are encouraging (Meier et al., 2024). Running LES for all global power plants is obviously not practical. Our method does however provide detailed insights into the roles of chemistry and mixing on quantifying emissions using TROPOMI observations.

We will add towards the end of the manuscript:

> In general, our simulations provide new insights in the factors that are important for the interpretation of satellite-observed $NO_2$ plumes. Although it is impractical to run the model for each observed plume, it is likely that the main impacts can be parameterised in light-weight methods, like recently shown in Meier et al. (2024).

**References**

Ehhalt, D. H. and Rohrer, F.: Dependence of the OH concentration on solar UV, JOURNAL OF GEOPHYSICAL RESEARCH, 105, 3565–3571, , 2000.

Hakkarainen, J., Szelag, M. E., Ialongo, I., Retscher, C., Oda, T., and Crisp, D.: Analyzing nitrogen oxides to carbon dioxide emission ratios from space: A case study of Matimba Power Station in South Africa, Atmospheric Environment: X, 10, 100 110, `https://www.sciencedirect.com/science/article/pii/S2590162121000101`, 2021.

Lelieveld, J., Peters, W., Dentener, F. J., and Krol, M. C.: Stability of tropospheric hydroxyl chemistry, Journal of Geophysical Research, 107, 4715, , 2002.

Meier, S., Koene, E. F. M., Krol, M., Brunner, D., Damm, A., and Kuhlmann, G.: A light-weight NO2 to NOx conversion model for quantifying NOx emissions of point sources from NO2 satellite observations, EGUsphere [preprint], -, , 2024.

Valin, L. C., Russell, A. R., and Cohen, R. C.: Variations of OH radical in an urban plume inferred from NO2 column measurements, Geophysical Research Letters, 40, 1856–1860, , 2013.

[Figure]

Figure 2: $I_{s,NO_2,OH}$ (in percent) calculated over the entire domain and degraded to TROPOMI resolution. Values are calculated up to the height of the convective boundary layer. These boundary layer heights are respectively 2500 (JAE1), 2000 (JAE2), 1200 (BEL1), 1500 (BEL2), 1800 (LIP1), 1500 (LIP2), 1900 (MAT1), and 1850 m (MAT2).

[Figure]

Figure 3: NO$_2$ lifetime calculated over the entire domain and degraded to TROPOMI resolution. Values are calculated up to the height of the convective boundary layer. These boundary layer heights are respectively 2500 (JAE1), 2000 (JAE2), 1200 (BEL1), 1500 (BEL2), 1800 (LIP1), 1500 (LIP2), 1900 (MAT1), and 1850 m (MAT2).

[Figure]

Figure 4: NO$_x$ lifetime calculated over the entire domain and degraded to TROPOMI resolution. Values are calculated up to the height of the convective boundary layer. These boundary layer heights are respectively 2500 (JAE1), 2000 (JAE2), 1200 (BEL1), 1500 (BEL2), 1800 (LIP1), 1500 (LIP2), 1900 (MAT1), and 1850 m (MAT2).

[Figure]

Figure 5: NO$_x$ to NO$_2$ ratio calculated over the entire domain and degraded to TROPOMI resolution. Values are calculated up to the height of the convective boundary layer. These boundary layer heights are respectively 2500 (JAE1), 2000 (JAE2), 1200 (BEL1), 1500 (BEL2), 1800 (LIP1), 1500 (LIP2), 1900 (MAT1), and 1850 m (MAT2).